**Do Land Models Miss Key Soil Hydrological Processes Controlling Soil Moisture Memory?**

**Mohammad A. Farmani[1], Ali Behrangi[1,2], Aniket Gupta[1], Ahmad Tavakoly[3,4], Matthew Geheran[3], and Guo-Yue Niu[1]**

[1]Department of Hydrology and Atmospheric Sciences, University of Arizona, Tucson, AZ, USA,
[2]Department of Geosciences, University of Arizona, Tucson, AZ, USA,
[3]US Army Engineer Research and Development Center, Coastal and Hydraulics Laboratory, Vicksburg, MS, USA,
[4]Earth System Science Interdisciplinary Center, University of Maryland, College Park, MD, USA

Corresponding author: Mohammad Farmani, email: farmani@arizona.edu

Guo-Yue Niu, email: niug@arizona.edu

**Key Points:**

Van-Genuchten soil hydraulics improves long-term Soil Moisture Memory (SMM) of the topsoil.

Surface ponding enhances soil moisture memory in both topsoil and the root zone.

Representing preferential flow improves both short-term and long-term SMM in both the topsoil and root zone.

**Abstract**

Soil moisture memory (SMM), which refers to how long a perturbation in Soil Moisture (SM) can last, is critical for understanding climatic, hydrologic, and ecosystem interactions. Most land surface models (LSMs) tend to overestimate surface soil moisture and its persistency (or SMM), sustaining spuriously large soil surface evaporation during dry-down periods. We attempt to answer a question: Do LSMs miss or misrepresent key hydrological processes controlling SMM? We use a version of Noah-MP with advanced hydrology that explicitly represents preferential flow and surface ponding and provides optional schemes of soil hydraulics. We test the effects of these processes that are generally missed by most LSMs on SMM. We compare SMMs computed from various Noah-MP configurations against that derived from the Soil Moisture Active Passive (SMAP) Level 3 soil moisture and in-situ measurements from the International Soil Moisture Network (ISMN) from year 2015 to 2019 over the contiguous United States (CONUS). The results suggest that 1) soil hydraulics plays a dominant role, and the Van-Genuchten hydraulic scheme reduces the overestimation of the long-term surface SMM produced by the Brooks-Corey scheme, which is commonly used in LSMs; 2) explicitly representing surface ponding enhances SMM for both the surface layer and the root zone; and 3) representing preferential flow improves the overall representation of soil moisture dynamics. The combination of these missing schemes can significantly improve the long-term memory overestimation and short-term memory underestimation issues in LSMs. We suggest that LSMs for use in seasonal-to-subseasonal climate prediction should, at least, adopt the Van-Genuchten hydraulic scheme.

## Plain Language Summary

Land surface models (LSMs) represent the physical and bio-geochemical exchanges of mass and energy between surface and atmosphere. Such exchanges are extensively dependent on the surface soil water amount and its persistence. This study explores key hydrological processes that may be missed by LSMs but important for weather and climate predictions. Through virtual experiments with a state-of-the-art model, we found that soil hydraulics (representing how efficiently soil can hold/release water under varying pressure) is particularly effective in sustaining soil moisture. Additionally, we found that allowing water to pond on the soil surface helps improve the model's soil moisture persistency. Furthermore, enhanced soil permeability due to soil macropores also regulates the water movement hence improving the soil moisture persistency. Overall, future LSMs should refine the treatment of soil water retention capability and consider the effects of soil macropores and surface ponding to improve weather and seasonal climate predictions.

## 1. Introduction

Land surface models' (LSMs) efficacy in simulating climate feedback mechanisms critically depends on the soil water retention capacity and soil moisture persistency. Rainwater that rapidly infiltrates into deeper subsoil strata is unavailable to be returned to the atmosphere through evaporation, thereby preventing potential atmospheric feedback loops (McColl et al., 2019). The influence of soil moisture on climate predictions at seasonal-to-sub-seasonal (S2S) scales is well-recognized due to its role in the exchange of surface energy and water fluxes with the atmosphere (Koster et al., 2002; Randal D. Koster et al., 2009; Koster et al., 2010; Koster & Suarez, 2001). Water stored in soil and aquifers, which variably persists from seasons to years, is known to affect precipitation variability (Koster & Suarez, 1999, 2001). This impact is particularly pronounced in regions transitioning from dry to wet conditions, where evapotranspiration (ET) is highly sensitive to soil moisture levels (Zhichang Guo et al., 2006; Koster et al., 2004; Koster & Suarez, 2001; Seneviratne, Koster, et al., 2006). While the nature and scale of soil moisture-precipitation feedback are still being debated (Findell et al., 2011; Taylor et al., 2013), numerous studies have emphasized the importance of soil moisture initialization and its persistency for accurate climate predictions (Dirmeyer, 2011; Mei & Wang, 2012; Shellito et al., 2016; Tuttle & Salvucci, 2016; Hossein Yousefi Sohi et al., 2024; Zebarjadian et al., 2024; Zeng et al., 2010). The degree of soil moisture-precipitation coupling widely varies across different climate models (Koster et al., 2004; Koster & Suarez, 1999; Moghisi et al., 2024; Seneviratne & Koster, 2012; Taylor et al., 2013), and discrepancies in the modeled soil moisture by LSMs for climate modeling are notable (Boone, 2004; Souri et al., 2024).

Refinement of soil moisture-precipitation feedback in LSMs is hindered by the lack of large-scale observational data, challenging the improvement and validation of model simulations (Koster et al., 2010; Koster & P. Mahanama, 2012; Koster & Suarez, 1999, 2001; Seneviratne & Koster, 2012). This shortfall highlights the necessity for more detailed representations of land-atmosphere feedback mechanisms that are crucial for extreme weather event predictions, yet are typically parameterized rather than explicitly resolved in models (McColl et al., 2019; Pastorello et al., 2020). Integrating extensive observational data is vital for simulating the intricacies of climate and weather and improving model predictive skill (Koster et al., 2017; R. D. Koster et al., 2009; McColl et al., 2019; Shellito et al., 2018). Recent advancements in remote sensing observations have enabled analyses of interactions between near-surface soil and the atmosphere. Nonetheless, the paucity of root zone data complicates the investigation of deep soil dynamics. Numerous studies have utilized satellite soil moisture products to evaluate and refine models, focusing on the spatial and temporal patterns of soil moisture variability (Randal D. Koster et al., 2009; Yang et al., 2020). In particular, the Soil Moisture Active Passive (SMAP) mission has been extensively employed to assess model performance (McColl, Alemohammad, et al., 2017; McColl et al., 2019; McColl, Wang, et al., 2017; Shellito et al., 2016; Shellito et al., 2018).

The concept of Soil Moisture Memory (SMM)— the duration required for a perturbation, such as rainfall, to dissipate—becomes essential for understanding the land-atmosphere interactions. SMM encapsulates the temporal variations of soil moisture, reflecting the exchange of fluxes between land and atmosphere. Therefore, SMM is an important metric for evaluating LSMs, since one of their functions is to provide surface flux exchanges and boundary conditions for atmospheric models (Z. Guo et al., 2006; Koster et al., 2004; Randal D. Koster et al., 2009; R. D.

Koster et al., 2009; Seneviratne, Koster, et al., 2006). SMM also facilitates the comparison of how
quickly soil loses water between observations and various models, providing insights into the
mechanisms within LSMs and their hydrometeorological responses. Moreover, analyzing SMM
can yield valuable data on the configurations and hydrological parameterizations of specific LSMs,
thus improving our understanding of how different configurations impact model performance,
particularly in soil moisture representation. For instance, Shellito et al. (2018) measured the drying
rate of surface soil moisture, which they considered as soil moisture memory, using SMAP data
and the Noah LSM during the initial 1.8 years following SMAP's launch. They concluded that
Noah shows a slower drying rate and a longer surface SMM compared with SMAP, due likely to
the too strong soil water suction represented by Noah.
Determining SMM is not straightforward due to the variety of calculation methods proposed by
researchers (Ghannam et al., 2016; Katul et al., 2007; Koster et al., 2004; Koster et al., 2002;
Randal D. Koster et al., 2009; Koster & Suarez, 1999, 2001; Mao et al., 2020; McColl,
Alemohammad, et al., 2017; McColl et al., 2019; McColl, Wang, et al., 2017; Seneviratne, Koster,
et al., 2006; Shellito et al., 2016), each introducing its own level of uncertainty. Traditionally, soil
moisture has been conceptualized as a red noise process, forming the basis for SMM calculations
(T. L. Delworth & Manabe, 1988). This approach has led to the definition of SMM as the e-folding
autocorrelation timescale within such a process (Delworth & Manabe, 1989). SMM has also been
characterized using various other autocorrelation-based methods, such as the integral timescale
(Ghannam et al., 2016; Nakai et al., 2014), soil moisture variance spectrum (Katul et al., 2007;
Nakai et al., 2014), and the constant time lag autocorrelation (Koster & Suarez, 2001; Seneviratne,
Lüthi, et al., 2006). Traditionally, these models were applied to monthly datasets. However, this
approach risks overlooking dynamic processes governed by limitations in water and energy
(Mccoll et al., 2019). Consequently, there has been a shift away from their use towards recent high-
resolution observational and modeling data. Therefore, there is a need for further research to refine
SMM measurement that can then be used as a benchmark for assessing LSMs (Mccoll et al., 2019).
McColl et al. (2019) categorized soil water loss into two main categories: water-limited (long-
term) and energy-limited (short-term). The energy-limited regime is a process where water loss is
constrained by available energy and lasts from hours to a few days. In contrast, the water-limited
regime is a process where water loss depends on the available water and spans longer periods, such
as weeks, months, and seasons. McColl et al. (2019) specified that ET and drainage are the main
controllers of long-term and short-term memories, respectively. Utilizing a two-year dataset from
the SMAP mission and simulations from the Goddard Earth Observing System Model, Version 5
(GEOS-5), McColl et al. (2019) conducted a global analysis under various climatic and land
conditions. Their analysis revealed that GEOS-5 tends to overpredict the duration of water-limited
memory and underpredicts energy-limited memory compared to SMM inferred from SMAP data,
while the results were not affected by the SMAP sampling frequency of 3 days. Building on this,
He et al. (2023) employed the hybrid memory approach proposed by McColl et al. (2019) to assess
the hydrometeorological response of various LSMs, including GLDAS-CLSM, GLDAS-Noah,
MERRA2, NCEP, ERA5, and JRA55, against SMAP observations for 2015 – 2020. The authors
observed that LSMs generally overestimate memory in water-limited regime and significantly
underestimate it in energy-limited regime. Moreover, their study suggested that discrepancies in
SMM representation within LSMs are more attributable to the physical processes incorporated
rather than factors such as soil layer thickness or the nature of model simulations (online/offline)
(He et al., 2023).
A recent review on SMM identified the soil properties and processes as an important controlling
factor of SMM in addition to atmospheric forcings and land use and management for future studies
to examine the fundamental mechanisms of SMM emergence (Rahmati et al., 2024). Based on the
works of McColl et al. (2019) and He et al. (2023), this study aims to examine the impacts of key
soil hydrological processes and soil hydraulics on SMM that may be missed in most LSMs. Current
LSMs may be not enough to address the uncertainties of SMM estinmates for incomplete
representations of key hydrological processes controlling SMM and uncertainties in soil hydraulic
parameters (Rahmati et al., 2024). As such, we use a version of Noah-MP with advanced
hydrological representations of preferential flow, surface ponding, runoff of surface ponded water
(infilration excess runoff), and lateral infiltration, etc. (Niu et al., 2024). We conduct model
experiments with various soil hydraulic parametrizations of those by Brooks and Corey (1964) and
Van-Genuchten (1980), preferential flow, and surface ponding depth. Our analysis investigates
the impact of these configurations on soil moisture persistency across ET regimes and drainage,
so that it can provide insight into these missing physical processes affecting SMM. By comparing
SMM produced by various settings of Noah-MP with SMAP Level 3 data and ISMN observations
from 2015 to 2019 over the CONUS, we seek to identify key processes and soil hydraulic schemes
controlling SMM and thus provide guidance for future developments of LSMs (e.g., reduce the
prevalent SMM overestimations in LSMs).

## 2. Materials and Methods

SMM denotes the duration required for a perturbation to dissipate, or the period from the start to
the end of a perturbation. For instance, following precipitation, the change in near-surface soil
moisture marks the beginning of the perturbation. This excess moisture gradually diminishes due
to flux exchange or percolation to deeper soil layers. The moisture level of soil plays a critical
role in influencing water loss patterns. Following rainfall, the upper layer of soil initially holds
more moisture than its field capacity ($\theta_{fc}$), causing runoff and drainage (see Figure 1a).
Subsequently, as the soil gradually dries, its moisture content reduces to a range between $\theta_{fc}$ and
the critical threshold ($\theta_c$). This phase leads to consistent water loss at the maximum ET rate, known
as Stage-I ET. As this process continues, the soil moisture falls below $\theta_c$ (Figure 1a), at which
stage ET becomes limited by the available water, termed Stage-II ET or ET at water-limited regime
(illustrated in Figure 1a & b). Ultimately, when the soil moisture drops below the wilting point
($\theta_w$), water no longer leaves the soil. Therefore, the whole process of water loss depends on the
soil's moisture level and falls into two main types: energy-limited including unresolved drainage,
and Stage-I ET, and water-limited including Stage-II ET (Figure 1b) (Mccoll et al., 2019; He et al.
2023). Energy-limited, green strips, and water-limited regimes, dotted-lines, are shown in soil
moisture times series at the Tonzi Ranch station (Figure 1c).

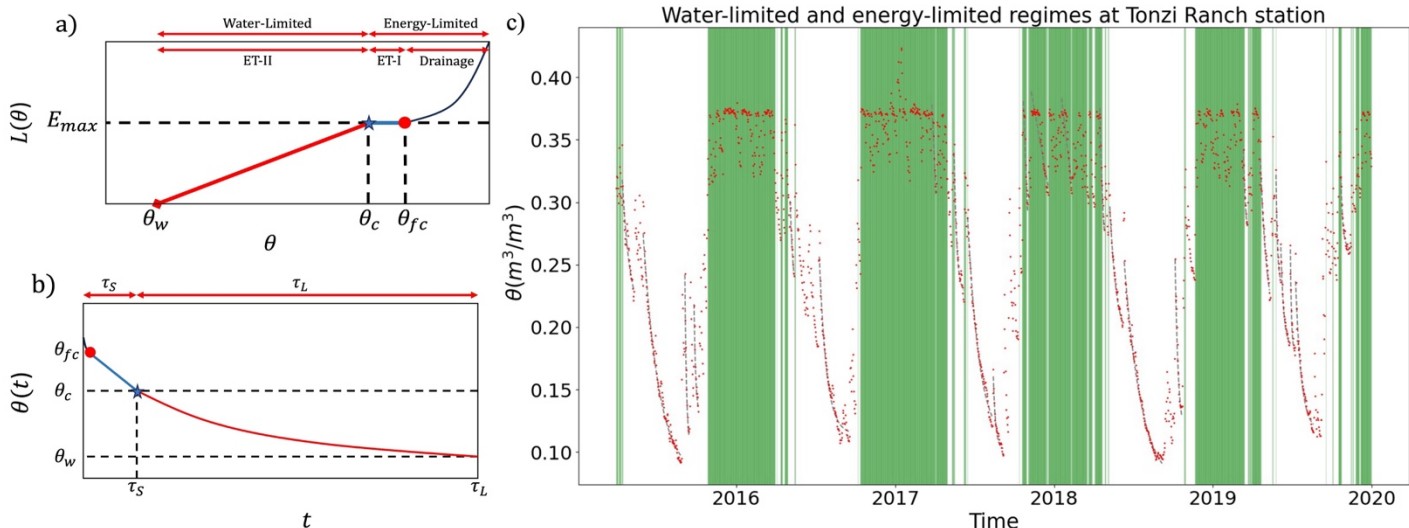

Figure 1 Schematic diagrams of (a) surface water loss process and (b) soil moisture memory at different soil moisture regimes [adapted from (McColl, Wang, et al., 2017b)]. Note that the x-axis in (a) refers to soil moisture ($m^3m^{-3}$), and y-axis refers to surface water loss rate, $L(\theta)$ (mm/s); $E_{max}$ is the maximum evaporation rate (mm/s). In (b), x-axis refers to time (e.g., days) and y-axis to SM content ($m^3m^{-3}$). Panel (c) shows the SM time series for the Tonzi Ranch station, with green periods indicating energy-limited regime and dotted lines representing water-limited regime. $\theta_w$, $\theta_c$ and $\theta_{fc}$ refer to the wilting point, critical point, and field capacity, respectively.


### 2.1. Soil Moisture Memory of Water-Limited Regime ($\tau_L$) and Energy-Limited Regime ($\tau_S$)

McColl et al. (2019) considered the SMM concept as it relates to two regimes: a) the memory of
water-limited regime ($\tau_L$), specified by 'L' abbreviation of Long-term, b) the memory of energy-
limited regime ($\tau_S$), specified by 'S' abbreviation of Short-term. Their model incorporates a
deterministic equation to represent water-limited processes during soil moisture drydown periods.
However, energy-limited processes occur over shorter timescales and present a challenge for
current satellite technologies to provide precise observations. McColl et al. (2019) highlighted that
drainage is not a completely resolved process by satellite observations. To address this gap,
McColl et al. (2019) proposed a stochastic equation to capture the unresolved nature of energy-
limited processes.
The hybrid model is formulated by McColl et al. (2019) as follows:

$$\frac{d\theta(t)}{dt} = \begin{cases} \dfrac{-\theta(t) - \theta_w}{\tau_L}, P = 0 \\ \dfrac{-\theta(t) - \overline{\theta}}{\tau_S} + \varepsilon(t), P > 0 \end{cases} \quad (1)$$

where, $\theta$ is the volumetric soil moisture, P indicates precipitation, $\theta_w$ is the minimum soil moisture, $\overline{\theta}$ is the time-averaged SM, and $\varepsilon(t)$ is a random variable with a mean of zero. $\tau_L$ and $\tau_S$ are SMM for the water-limited and energy-limited regimes, respectively. McColl et al. (2019) solved these equations, demonstrating that the memories can be expressed as:

$$\theta(t) = \Delta\theta \, exp\left(\frac{-t}{\tau_L}\right) + \theta_w P = 0 \tag{2}$$

$$\tau_S = \frac{-\frac{\Delta t}{2}}{log} \tag{3}$$

$\Delta\theta$ represents the soil moisture changes during drydown, $\Delta t$ is the temporal resolution of the soil moisture data, $\alpha$ is the precipitation intensity, $\Delta z$ is soil layer thickness, and $\overline{\Delta\theta_+} = \theta(t) - \theta(t-\Delta t)$ represents a positive increment in soil moisture. (McColl, Alemohammad, et al., 2017) defined $\frac{\Delta z\,\overline{[\Delta\theta_+]}}{\alpha}$ as stored fraction of precipitation, indicating the average proportion of water that still exists in soil layer $\Delta t$ days after rainfall. McColl et al. (2019) declared that the short-term memory in their hybrid model is dominated by drainage when the sampling is relatively high (as in the case of SMAP's sampling frequency of 3 days). This approach and its rationale are further elaborated in (McColl, Alemohammad, et al., 2017) and McColl et al. (2019).

In the analysis of water-limited memory, we fitted Equation 2 to the soil moisture time series during specific drydown intervals. Then, $\tau_L$ was extracted as a parameter from the fitting curve (black dotted lines in Figure 1c). In contrast, short-term memory was determined directly using Equation 3, as indicated by the green periods in Figure 1c. Further information about the criteria for calculating memories can be found in McColl et al. (2019).

## 2.2. Description of Datasets

We use high-resolution atmospheric forcing datasets to drive the Noah-MP LSM. This model is set up to simulate soil moisture dynamics, featuring advanced infiltration and water retention processes. Additionally, it includes a precise parameterization for ponding depth. This setup facilitated five distinct experiments. Then, we used surface and root zone soil moisture data derived from the Noah-MP experiments, SMAP Level 3 surface soil moisture measurements, and root zone soil moisture measurements from the International Soil Moisture Network (ISMN) to calculate the hybrid SMM. The rest of this section describes in detail the forcing and observational datasets, the Noah-MP LSM configurations, the employed infiltration and water retention schemes, and the ponding depth threshold criterion.

### 2.2.1 Atmospheric Forcing, Soil and Vegetation Parameters

For modeling purposes, this study utilized the North American Data Assimilation System Phase 2 (NLDAS-2) near-surface meteorological data at an hourly interval and 0.125° spatial resolution. This dataset encompasses a range of variables including air temperature, specific humidity, wind speed, surface pressure, shortwave and longwave radiation, and precipitation (Xia et al., 2012). We also used precipitation data from the Integrated Multi-satellite Retrievals for Global Precipitation Measurement (IMERG-Final) dataset (Huffman et al., 2020; Jawad et al., 2024; H. Yousefi Sohi et al., 2024), which offers half-hourly measurements across a 0.1° grid extending from 60°S to 60°N. Subsequently, the IMERG-Final data were mapped to the 0.125° resolution of NLDAS-2 using bilinear interpolation. These precipitation data sources were integrated into the short-term SMM computation process. To integrate the IMERG precipitation product into the model, we modified the forcing component of the Noah-MP code. Specifically, an average of NLDAS-2 and IMERG precipitation was employed when NLDAS-2 reported negative precipitation values, which was particularly significant in coastal regions. This adjustment enhanced the accuracy of precipitation inputs, contributing to more reliable simulations in these areas.

To ascertain soil and vegetation parameters, the hybrid State Soil Geographic Database (STATSGO) with a 1-km resolution and the United States Geological Survey's (USGS) 24-category vegetation classification were employed. The datasets were aggregated to align with a 0.125° resolution, which is consistent with the NLDAS-2 forcing data. This process included determining the dominant soil and vegetation types for each grid cell. Subsequently, the lookup tables within the Noah-MP model (Niu et al., 2020) were used to assign the relevant parameters to the corresponding soil and vegetation categories.

### 2.2.2 SMAP L3 Surface Soil Moisture

Since its successful deployment on January 31, 2015, the Soil Moisture Active Passive (SMAP) observatory has consistently provided global volumetric soil moisture estimates every two or three days (Entekhabi et al., 2010). Its onboard radiometer, operating in the L-band frequency of the microwave spectrum, senses the top five centimeters of the soil column. In this study, we selected the SMAP Level 3 morning overpass due to the greater likelihood of air and surface temperature equilibrium during these hours, a critical condition for the SMAP retrieval algorithm. The L3 SMAP data used here span from 2015 to 2020, have a spatial resolution of 9 kilometers and are instrumental in calculating SMM across the Continental United States (CONUS).

In line with established methodologies from previous research (He et al., 2023; Mccoll et al., 2019), a quality control protocol was deemed necessary to refine soil moisture data in regions affected by dense vegetation, bodies of water, and permafrost, thereby mitigating noise present in satellite measurements (He et al., 2023; Mccoll et al., 2019; McColl, McColl, Wang, et al., 2017). However, this study is conducted to determine SMM to deepen our knowledge of physical processes and to get closer to optimal soil hydraulic parametrizations within Noah-MP. This is achieved through a comparative analysis of SMM derived from SMAP and Noah-MP datasets. Given that a specific parametrization within Noah-MP has a pronounced impact on the eastern region of the Continental United States (CONUS)—a region that also corresponds with a significant portion of SMAP's low-quality data—we chose not to filter SMAP data to fully capture the parametrization effects within our study's geographical focus. This approach was intended to maintain consistency across figures and enhance the presentation of our findings. Furthermore, our

objective is to showcase the physical process involved in SMM, rather than focusing on model
accuracy in comparison with SMAP data. Note that the SMM maps from McColl et al (2019) and
He et al (2023) demonstrated the effect of removing SMAP low-quality data, and hence we did
not include the map of locations with high-quality SMAP data. Given that the surface water
balance is sensitive to the temporal resolution of the analyzed surface soil moisture data, the SMAP
L3 soil moisture data are resampled to achieve a consistent sampling frequency of one per three
days at each pixel (He et al., 2023; McColl, Wang, et al., 2017).

### 2.2.3 International Soil Moisture Network (ISMN)

In evaluating the Noah-MP model's parametrization for the root zone soil moisture, SMM is
computed using both the model's outputs and in situ observations across the CONUS. We obtained
the in situ soil moisture data from the International Soil Moisture Network (ISMN) portal (Dorigo
et al., 2011), which compiles quality-controlled measurements from various sensors across
multiple networks, Figure *2*. We exclude stations with less than 90% of their data rated as "good"
quality. Despite the diversity of sensor types within ISMN, its stringent quality assurance protocols
suggests that it is a reliable benchmark for validating soil moisture products(Colliander et al., 2017;
Shellito et al., 2016). For the representation of root zone soil moisture, we select only the data from
the top 1 meter of soil flagged as "good" quality. These measurements are averaged, i.e., hourly
data aggregated to daily means, and the daily time series are used to compute both long-term and
short-term SMM.

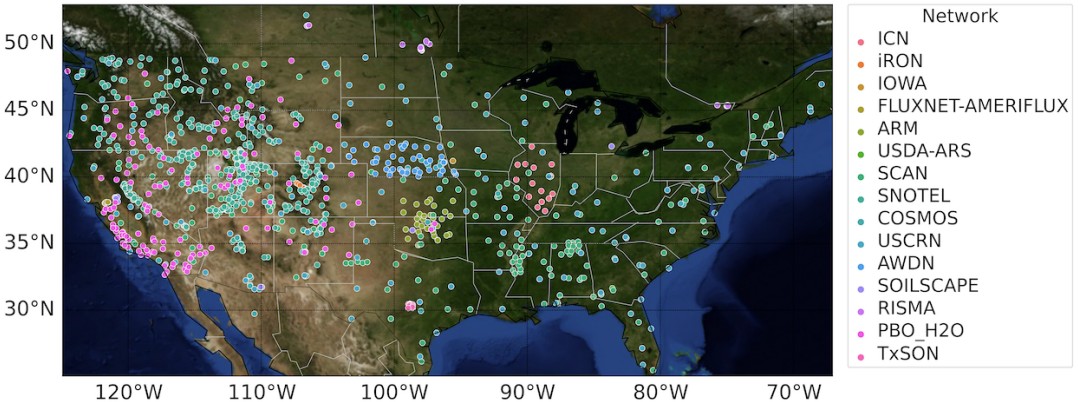

Figure 2 ISMN in-situ locations and networks over CONUS.


### 2.3 Noah-MP with Advanced Soil Hydrology

In this study, we choose Noah-MP (Niu et al., 2024; Niu et al., 2011; Yang et al., 2011) for its
extensive use within the Weather Research and Forecasting (WRF) model, the Unified Forecast
System (UFS) for weather and short-term climate projections, and the National Water Model
(NWM) for streamflow and water resource forecasting. The "semi-tile" sub-grid methodology of
Noah-MP enables detailed calculation of surface energy and fluxes, differentiating effectively
between bare and vegetated terrains to precisely compute variables such as latent and sensible heat
fluxes (Agnihotri et al., 2023).

The Noah-MP version used in this study includes additional developments in plant hydraulics that
explicitly represent plant water storage supplied by root water uptake driven by the hydraulic
gradient between the soil and roots (Niu et al., 2020) and advanced soil hydrology that solves
mixed-form Richards' equation and thus explicitly represents surface ponding, infiltration of
surface ponded water, and preferential flow (Niu et al, 2024). As such, current Noah-MP accounts
for water flow driven by the hydraulic gradients from the soil to the vegetation canopy to meet the
plant transpiration demand. It also accounts for subgrid variability in infiltration capacity through
a fractional area of preferential flow pathways caused by soil macropores in the fields. A detailed
description of the underlying physical mechanisms for the schemes used in this study can be found
in Niu et al, (2024), also a brief description of equations and parameters is included in supporting
material.

**The Mixed-Form Richards' Equation:** Most LSMs solve the mass-based (or $\theta$-based) Richards'
Equation (RE) for unsaturated soils(Chen & Dudhia, 2001; Oleson et al., 2010) and thus are not
adequate to represent saturated conditions, e.g., surface ponding and groundwater dynamics. The
current Noah-MP adopts the methodology of (Celia et al., 1990) to solve the mass-pressure ($\theta$-h)
mixed-form RE (MF). The new solver solves pressure head, h, and conserves mass due to the mass
($\theta$) constraint. To achieve a more accurate solution of *h* and mass balance, the solver takes an
adaptive time stepping scheme.
Surface ponding occurs when the pressure head of the surface layer is greater than the air entry
pressure, and the upper boundary condition (BC) shifts from flux BC to head BC following
Paniconi (1994). Infiltration-excess runoff occurs when the surface ponding depth, $H_{top}$, surpasses
a predefined threshold, $H_{top,max}$, at which the surface ponded water at local depressions of a model
grid starts to be connected and runs off. The model extends its vertical domain to the bedrock depth
(Pelletier et al., 2016) at which the lower BC is set up as a zero-flux BC. Groundwater discharge
is simply represented using the TOPMODEL concept as a function of water table depth, which is
determined by the modeled pressure head, which is interpolated between saturated zone and its
overlying unsaturated zone.
**Optional Soil Hydraulics Schemes:** The current Noah-MP provides optional hydraulics schemes
of the Van Genuchten-Mualem (VGM) and the Brooks-Corey with Clapp-Hornberger (BC/CH)
parameters. To facilitate quicker convergence, particularly near saturation, we smoothed the
BC/CH water retention curve using a polynomial function following (Bisht et al., 2018).
**Representing Preferential Flow:** To represent preferential flow, current Noah-MP adopts a dual-
permeability model (DPM) approach, partitioning the model grid into two domains: one
representing rapid flow with reduced suction head (macropores) and the other for slower matrix
flow, following Šimůnek & van Genuchten, (2008) and Gerke and van Genuchten (1993a,b, 1996)
(Gerke & van Genuchten, 1993a, 1993b; Gerke & van Genuchten, 1996; Šimůnek & Van
Genuchten, 2008). This approach represents subgrid variability in infiltration capacity through a
fractional area of soil macropores in the fields, $F_a$, (or volumetric fraction of macropores). DPM
also represents water transfer between the two pore domains, which can be either be positive
("lateral infiltration" during rainy days) or negative (diffusion from micropores to drier
macropores). It also accounts for lateral movement of surface ponded water from the matrix to
macropore domains at the soil surface. The aggregated water content ($\theta$) and vertical water flux
(q) for a grid cell are given by θ = $F_a$ θ$_a$ + (1−$F_a$) θ$_i$, and q = $F_a$ q$_a$ + (1−$F_a$) q$_i$, respectively,
where *q* denotes a water flux and the subscripts a and i respectively indicate macropore and
micropore domains. This approach also extends to other water fluxes, such as direct evaporation
from soil surface, $E_{soil}$, and groundwater recharge.

Table 1 Noah-MP Options used in this study.

| Process | Options | Schemes |
|---|---|---|
| Dynamic vegetation | DVEG = 2 | Dynamic vegetation |
| Canopy stomatal resistance | OPT_CRS = 1 | Ball-Berry type |
| Moisture factor for stomatal resistance | OPT_BTR = 1 | Plant water stress |
| Runoff and groundwater | OPT_RUN = 1 | TOPMODEL with groundwater |
| Surface layer exchange coefficient | OPT_SFC = 1 | Monin-Obukhov similarity theory (MOST) |
| Radiation transfer | OPT_RAD = 1 | Modified two-stream |
| Ground snow surface albedo | OPT_ALB = 3 | Two-stream radiation scheme (Wang et al., 2022) |
| Precipitation partitioning | OPT_SNF = 5 | Wet bulb temperature (Wang et al., 2019) |
| Lower boundary condition for soil temperature | OPT_TBOT = 2 | 2-m air temperature climatology at 8m |
| Snow/soil temperature time scheme | OPT_STC = 1 | Semi-implicit |
| Surface evaporation resistance | OPT_RSF = 1 | Sakaguchi and Zeng (2009) |
| Root profile | OPT_ROOT = 1 | Dynamic root (Niu et al., 2020) |

**2.4 Model Experiments**

We conducted five experiments using the current Noah-MP driven by the hourly NLDAS-2 forcing
data at a spatial resolution of 0.125 degree, starting with the same uniform initial conditions—
namely, soil moisture at 0.3 m3m–3 and soil temperature at 287K—spanning 2014 to 2019 for six
iterations. The initial five iterations were dedicated to the model's spin-up phase, and the resulting
surface and root zone soil moisture from the last iteration were used for SMM analysis. Parameters
were adopted per the updates by Niu et al. (2020), with adjustments to the dynamic vegetation
module to align with Moderate Resolution Imaging Spectroradiometer (MODIS) leaf area index
observations. This study refrained from parameter calibration related to dual-domain schemes for
preferential flow (Šimůnek & Van Genuchten, 2008) and ponding depth.

The five experiments are conducted with Noah-MP configurations with different water retention
and infiltration schemes. Table 1 lists optional schemes that were the same for all these
experiments. for other processes, including surface layer turbulent exchange, radiation transfer,
phase changes between snow and rain, and the permeability of frozen soil. For this study, we
selected only those schemes that have a direct impact on the simulation of soil moisture dynamics
(as detailed in Table *2*). All these experiments are set with the same number of soil layers, which
vary spatially from 5 – 15 vertical layers with fixed layer thicknesses: $\Delta z_i$ =0.05, 0.3, 0.6, 1.0, 2.0,
2.0, 4.0, 4.0, 5.0, 5.0, 5.0, 5.0, 5.0, 5.0, and 5.0 m down to 49.0 m to match the maximum bedrock
depth data of Pelletier et al. (2016) with a minimum bedrock depth of 4.0 m. The model was
customized using a combination of three soil moisture solver variants, two soil hydraulics schemes,
and two ponding depth thresholds.
To explore the influence of surface ponding on SMM, we designed two distinct experimental
conditions. The first condition, designated as MF_VGM0, excluded the ponding effect by setting
$H_{top,max}$ to 0 mm. Conversely, the second condition, identified as MF_VGM200, incorporated a
significant ponding depth of 200 mm. Both conditions utilized the mixed-form RE solver alongside
the Van-Genuchten (VGM) model (refer to Table *2*). Furthermore, we conducted comparative
analyses to assess the role of soil hydraulic properties by conducting experiments with the Brooks-
Corey/Clapp-Hornberger (BC/CH) model (MF_CH) and the VGM model (MF_VGM), each with
a ponding depth threshold of $H_{top,max}$ = 50 mm.
An additional experiment employs the Dual-Permeability model (DPM) within the VGM
framework, maintaining the same ponding threshold of $H_{top,max}$ = 50 mm, referred to as
DPM_VGM (see Table 2). The comparison of DPM_VGM with the MF_VGM setup aimed to
shed light on the effects of preferential flow channels on soil moisture forecasting, and runoff
forecasting in future studies, thereby enhancing our comprehension of the complexities inherent
in hydrological modeling.
To define the macropore volume fraction, we used the modeled Soil Organic Matter (SOM), which
is computed from Noah-MP with a microbial-enzyme model(Zhang et al., 2014) prior to the major
experiments conducted in this study through a long-term (120 years) spin-up simulation from 1980
– 2019 driven by the NLDAS data. The modeled SOM shows a pattern of less SOM in wet regions
but more in arid regions due to more active microbial activities (decomposition and respiration) in
wetter regions. The resulting macropore volume fraction ranges from 0.05 – 0.15 changing with
spatially-varying SOM. While we conducted sensitivity analyses on key parameters such as the
ponding depth threshold and macropore fraction to identify ranges yielding realistic outcomes, we
acknowledge that further model development (building relationships with global high-resolution
DEM and soil data, e.g., SoilGrids250m (Poggio et al., 2021) are necessary to refine the
parameters.
Table 2 Model experiment configuration.

| Experiment ID | Models | $H_{top,max}$ (mm) | Soil Hydraulics |
|---|---|---|---|
| MF_VGM0 | Mixed Form RE | 0 | Van-Genuchten |
| MF_VGM200 | Mixed Form RE | 200 | Van-Genuchten |
| MF_CH | Mixed Form RE | 50 | Brooks-Corey/Clapp-Hornberger |
| MF_VGM | Mixed Form RE | 50 | Van-Genuchten |
| DPM_VGM | DPM | 50 | Van-Genuchten |


## 3. Results

In Sections 2.1 and 2.2 of our study, we focus on computing the SMM for both the surface (5 cm) and root zone (up to 1m) layers, respectively. This dual-layer analysis is fundamental to our experiments as it allows us to understand the differential impacts of various parameterizations on soil moisture. By comparing and analyzing the SMM values across these two distinct layers, we can identify specific physical processes that influence soil moisture dynamics. This comparative approach not only elucidates how these processes affect SMM but also helps in understanding the interaction between surface characteristics and subsurface moisture dynamics, which are critical for improving hydrological modeling and prediction.

### 3.1 Long- and Short-Term Soil Moisture Memory of the Surface Layer

Figure *3* illustrates the spatial distribution of median long-term memory, derived from the five-year soil moisture dataset. We also provide plots for the SMM spatial distributions to offer insights for each model experiments. However, it turns out that interpreting the fundamental mechanisms behind the distribution is very challenging regarding the spatial distributions of other controlling factors, e.g., climatic forcing, vegetation/soil type, elevation, slope angle/aspect (affecting solar radiation), which directly or indirectly controls actual ET and runoff as well as interactions between ET and soil moisture (Rahmati et al., 2024). As such, we focus on comparing the median SMM values across model scenarios to find the dominate hydrological processes controlling SMM, because the modeled distributions from the different experiments generally show the same shape, especially for the same hydraulics (e,g., VGM). Analysis of the SMAP data revealed that long-term memory ($\tau_L$) is significantly higher in the energy-limited and humid regions of the eastern US, and lower in the arid western regions. These findings are consistent with those of He et al. (2023) and McColl et al. (2019).

The MF_CH experiment displays a spatial pattern that contrasts with the SMAP data, with a longer memory in the arid western regions but a shorter memory in the wet northeastern regions (Figure 3a & 3b). This is likely caused by the faster drainage of topsoil water under the wetter conditions, whereas under the drier conditions, the spuriously stronger suction from the CH hydraulics sustain the surface soil moisture for a longer period. Further examination reveals that models using the Van-Genuchten scheme reflect SMAP's patterns. Specifically, the eastern regions display higher $\tau_L$ values, while the western regions show lower values (see Figure 3b-f). DMP_VGM demonstrates a shorter memory in the eastern CONUS compared to MF_VGM (refer to Figures 3c, d, and S1. VGM scenario with zero ponding depth shows a shorter memory compared with MF_VGM200 in the eastern CONUS (Figures 3e and f), where surface ponding happens more frequently and with a greater depth. Figure S2 shows a better match of data points with the agreement line in the DPM_VGM versus SMAP scatterplot. In contrast, the MF_CH versus SMAP scatterplot lacks this alignment with a correlation of –0.10. The correlation values have risen from –0.10 to 0.15 with VGM, a sign of progress, but they are still not strong.

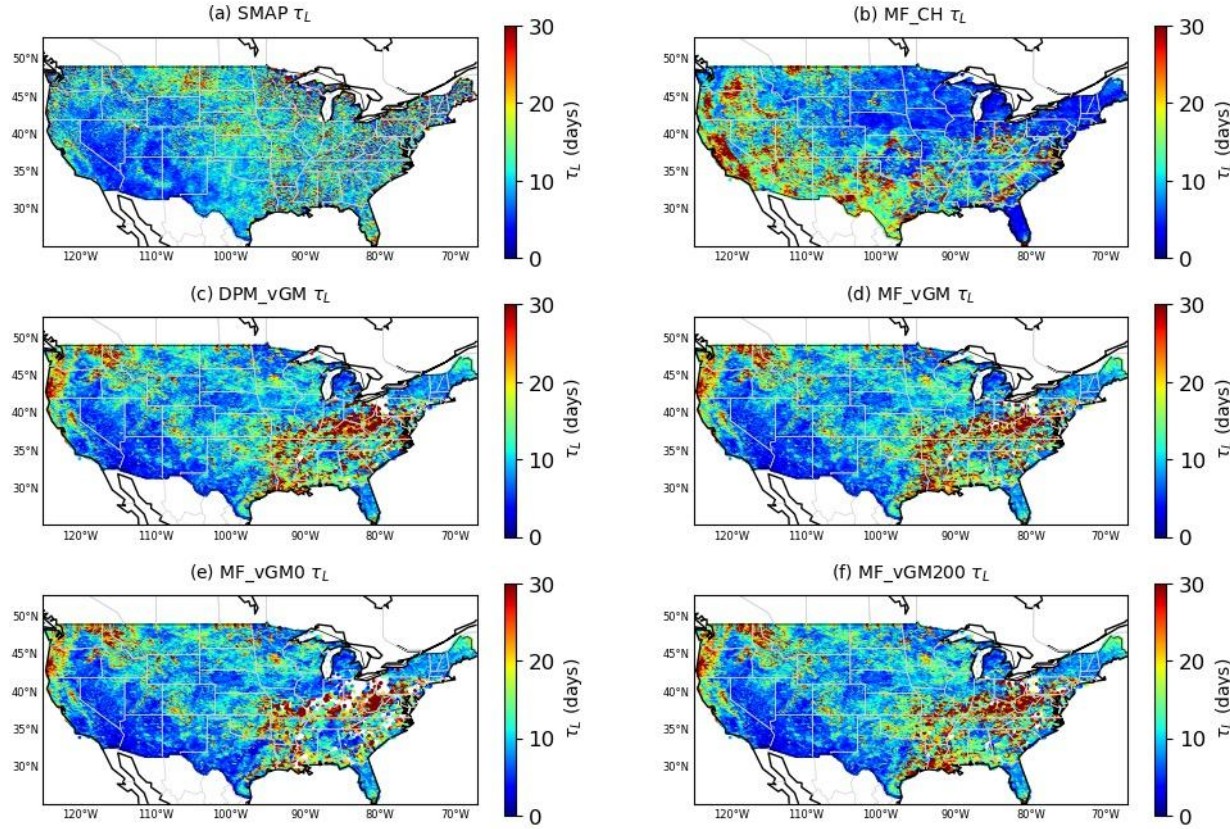

Figure 3. Long-term SMM derived from various datasets from 2015 – 2019 for soil surface layer: (a) SMAP; (b) MF_CH; (c) DMP_VGM; (d) MF_VGM; (e) MF_VGM0; and (f) MF_VGM200. SMM = Soil Moisture Memory

To assess the influence of plant water storage on SMAP soil moisture data and the resultant SMM,
we employed the MODIS NDVI to categorize the entire CONUS into wet (NDVI > 0.45) and dry
regions (NDVI < 0.45). In the dry areas (see Figure 4a), the probability distribution function (PDF)
of the surface SMM from MF_CH differs from that of SMAP and exhibits a higher median of
10.53 days compared to SMAP's 8.47 days (overestimation). Other model scenarios using van
Genuchten (VG) hydraulics, with an SMM median of around 8.6 days, show a distribution PDF
like SMAP. Note that the VGM scenarios effectively tackle the problem of long-term memory
overestimation, a point emphasized by He et al. (2023). This improvement is due to the refined
parametrization of physical processes within the VGM experiments.
In the wet regions with dense vegetation (Figure 4b), the SMM PDF of MF_CH (median of 8.03
days) significantly varies from the SMAP PDF (median of 10.71 days), showing an
underestimation of $\tau_L$. However, due to the strong effect of plant water storage on the SMAP's soil
moisture retrieval (commonly in the eastern CONUS), our focus here is on model sensitivity to
process representations rather than on model accuracy relative to SMAP data. Other models with
the van Genuchten (VG) scheme display greater variability among themselves in wet areas (Figure
4b) than in the dry region (Figure 4a). MF_VGM0 (with a zero ponding depth threshold) shows a
shorter long-term SMM, with a median of 10.72 days, compared to MF_VGM200 (with a 200 mm
threshold), with median of 12.05 days, and MF_VGM (with 50 mm ponding threshold), with a
median of 12.03. This suggests extra water inputs from the surface ponded water (MF_VGM200)
can help extend the surface SMM. Changing the ponding depth threshold from 50 mm (MF_VGM)
to 200 mm (MF_vGM200), has a marginal effect on $\tau_L$, suggesting that the response does not
proportionally increase with higher values. With the same 50 mm ponding threshold, DPM_VGM
produces a shorter SMM, with a median of 11.73 days, than MF_VGM, indicating that the effects
of faster water drainage of the topsoil water caused by the preferential flow (as represented by
DPM_VGM) can last longer.

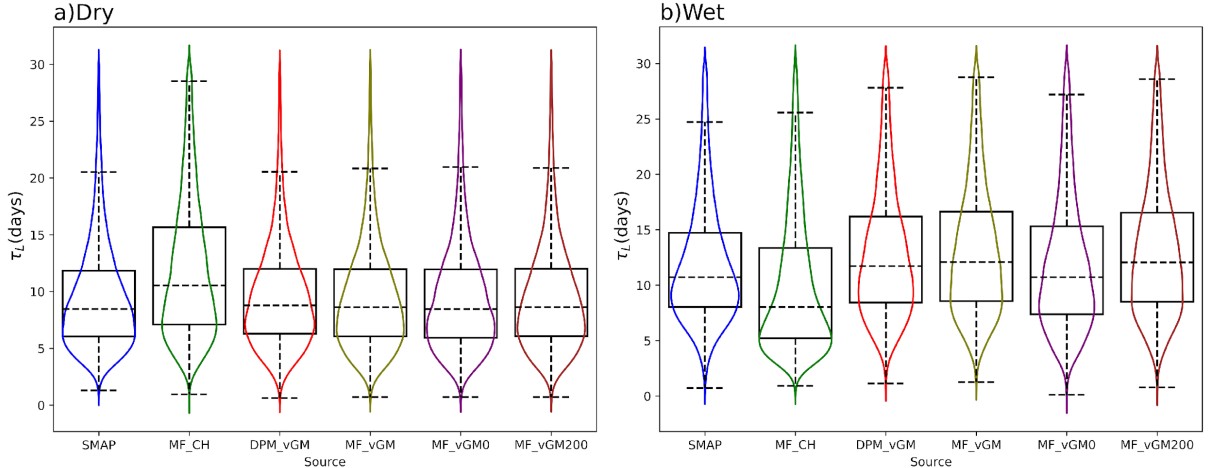

Figure 4 Violin plot of surface $\tau_L$ estimated from SMAP and Noah-MP scenarios for dry
regions with less vegetation (NDVI < 0.45) and wet regions with more vegetation (NDVI >
0.45).

For the short-term SMM, all the scenarios produce an overall spatial pattern similar to that of the
SMAP-derived $\tau_S$, showing a longer memory in the drier western US than in the wetter eastern
(Figure 5). However, MF_CH shows a shorter memory in the northwestern US than that derived
from SMAP (Figure 5a & b). MF_CH with a median of 1.9 days underestimates SMAP with a
median of 2.02 days, while VG scenarios have median $\tau_S$ around 2.09 days over dry regions. This
effectively rectifies the underestimation in short-term memory by LSMs, as reported in previous
studies (He et al., 2023). He et al. (2023) highlighted that most LSMs tend to underestimate $\tau_S$,
which is strongly affected by soil water drainage as specified by McColl et al. (2019). Note that
higher $\tau_S$ values indicate slow drainage, whereas lower values suggest faster drainage; this is
exemplified by Figure 5a, which exposes a more rapid drainage in the eastern CONUS in contrast
to the western. The incorporation of surface ponding and DPM (2.08 days) has shown less effects
on short-term memory than the soil hydraulics for the dry region (more macropores are available
in wet regions and hence DPM would have more effect in those regions). The introduction of
surface ponding (comparing MF_VGM0 (2.11 days) to MF_VGM200 (2.108 days) in Figure 5
and Figure 6) contributes to more persistent surface soil moisture and a bit faster drainage. The
pdf of SMM from all the VGM models more closely resembles the SMAP pdf in the western
United States than in the eastern part of the country due likely to that the SMAP soil moisture
retrieval may be affected by the plant water storage and thus the spatial variations in canopy
density.

For wet regions, MF_CH with a median of 1.26 days underestimate SMAP with a median of 1.56
days. DPM_VGM with faster drainage of surface soil water produces a median $\tau_s$ of 1.43, shorter
than does MF_VGM with a median of 1.48 days. The DPM model accelerates the drainage of
water from the topsoil. This effect is more significant in the eastern CONUS. As a result, it lowers
the short-term memory in areas where the soil has macropores.
The modeling results also indicate the long-term memory of the surface soil moisture is more
sensitive to the four VGM schemes in the wet regions (Figure 4b) than the short-term memory (
Figure 6b). This can be attributed to the differences in how topsoil water responds to surface
ponding and preferential flow as represented by the four VGM across different moisture regimes.
Under higher soil moisture conditions right after a rainfall event, the persistence of soil moisture
is mainly dominated by drainage of topsoil water to deeper soil, whereas at relatively lower soi
moisture, the long-term memory is more controlled by persistent water inputs from surface ponded
water and prolonged drainage by preferential flow. This also indicates that the effects infiltration
of surface ponded water and preferential flow can last longer up to more than 10 days. Under dry
conditions (Figure 4a and 6a), these hydrological processes become less important. However, the
soil water retention curves as represented by the CH and VG schemes play a more important role
under any conditions (Figure 4a and Figure 6a). Another possible reason can be the issue of time
scale. Short-term memory has values up to 5 days, and given the SMAP revisit time of 3 days,
generating values for intervals shorter than 3 days may challenge the validity of short-term
memory as a reliable measurement for soil drainage, as demonstrated by McColl et al. (2019).
Since we selected Noah-MP days corresponding to the SMAP revisit time, it is possible that the
effects of different VG parameterizations were diminished by this sampling. We suggest that other
measurements, such as streamflow and baseflow analysis, should be considered to better quantify
the effect of soil hydraulics on soil drainage. ~~Ji et al. (2023) demonstrated that high-resolution soil~~
~~datasets and model parameterizations can enhance these synergistic effects (Ji et al., 2023). This~~
~~variation in how local environmental conditions are represented likely explains the greater~~
~~variability observed in wet regions in Figure 4.~~

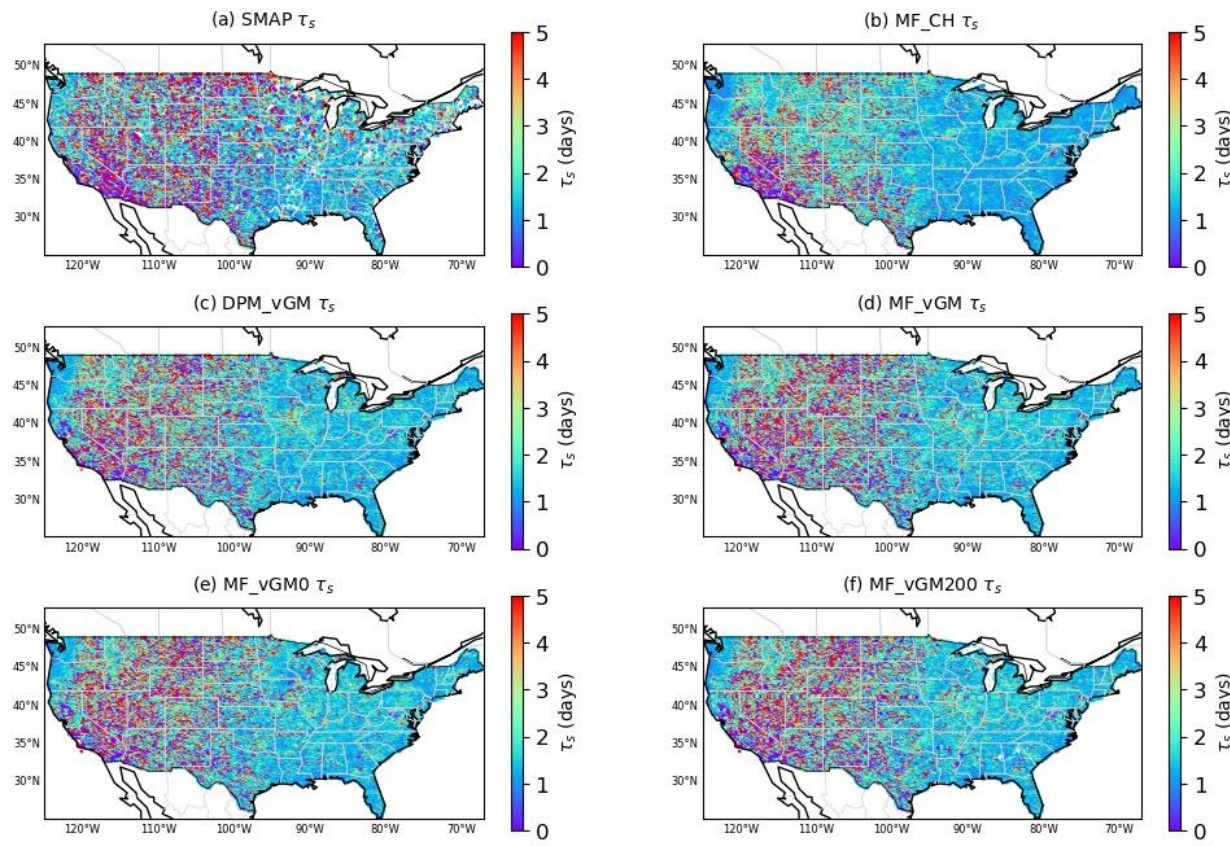

Figure 5 Short-term SMM derived from various datasets from 2015 – 2019 for soil surface layer: (a) SMAP; (b) MF_CH; (c) DMP_VGM; (d) MF_VGM; (e) MF_VGM0; and (f) MF_VGM200. SMM = Soil Moisture Memory.


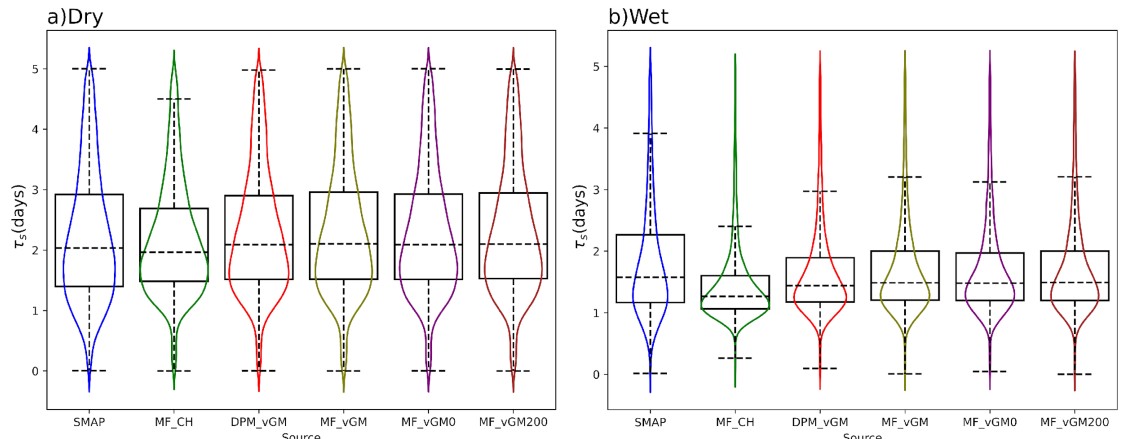

Figure 6 Same as Figure 4 for short-term memory.


## 3.2 Long- and Short-Term Soil Moisture Memory of the Root Zone Layers


We use the International Soil Moisture Network (ISMN) soil moisture dataset as the benchmark
and compute SMM at the ISMN stations as illustrated in Figure 2. We compute the long-term
SMM across 654 sites within CONUS for the period from 2015 – 2019. The median values of
these computations indicate that the root zone SMM (Figure 7 & Figure 9) is generally higher than
the surface SMM (Figure 3 & Figure *5*). Analysis of ISMN data reveals that the root zone $\tau_L$ (Figure
7) generally exceeds surface $\tau_L$ (Figure 3), particularly longer in the western US. Some eastern
locations also exhibit longer $\tau_L$, whereas the central region demonstrates lower values.
MF_CH produces a shorter root-zone $\tau_L$ across nearly all the sites in CONUS (Figure 7 & Figure
*8*). The Van-Genuchten scheme mirrors the ISMN-derived $\tau_L$, albeit with slightly higher values
(Figure 7 & Figure 8). An increase in surface ponding depth raises the $\tau_L$. This is particularly true
in the eastern US, where surface ponding occurs more often, and its impact on soil moisture is
more substantial. Figures S3 and S4 illustrate this effect. Additionally, DMP_VGM (Figure 7c and
Figure 8) reduces the root-zone long-term SMM across most of CONUS relative to the other
models (Figure 7c, d, e, & f and Figure S3).

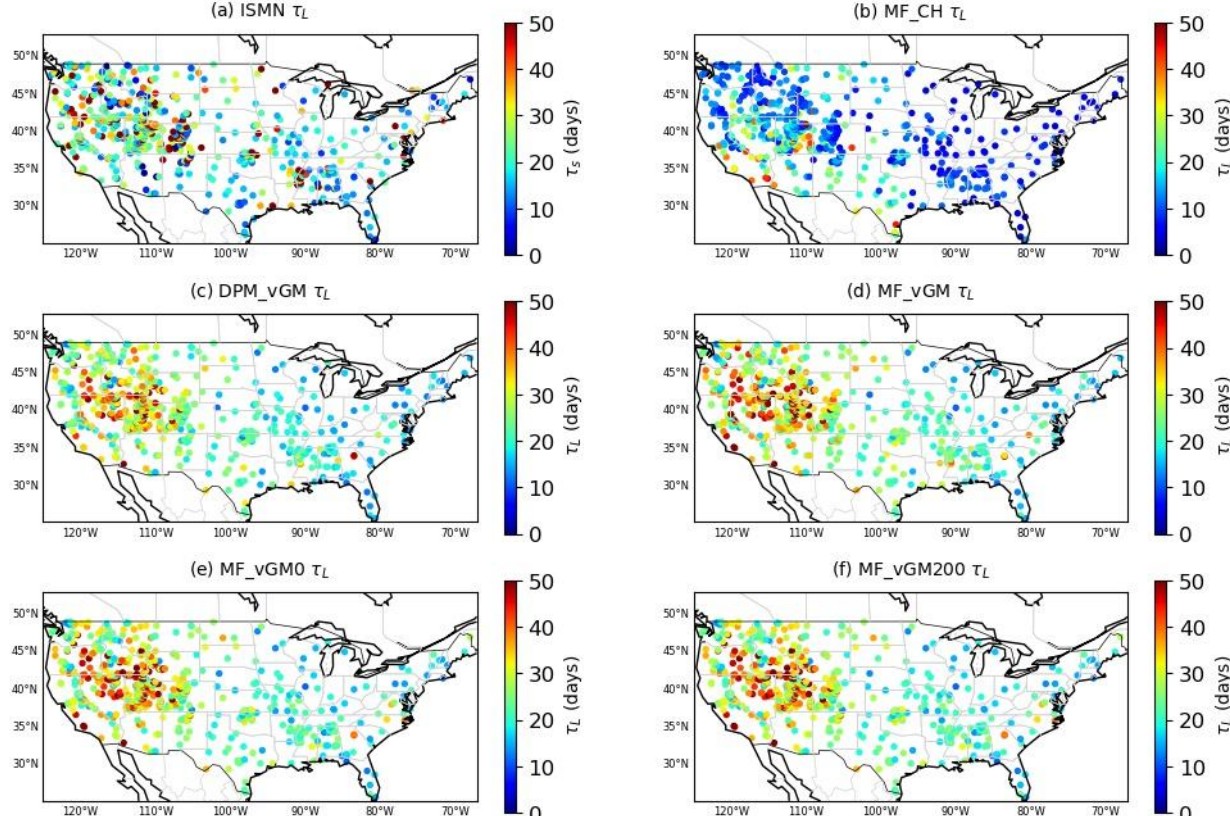

Figure 7 Long-term root-zone SMM derived from various datasets from 2015 – 2019: (a)
ISMN; (b) MF_CH; (c) DMP_VGM; (d) MF_VGM; (e) MF_VGM0; and (f) MF_VGM200.
SMM = Soil Moisture Memory.


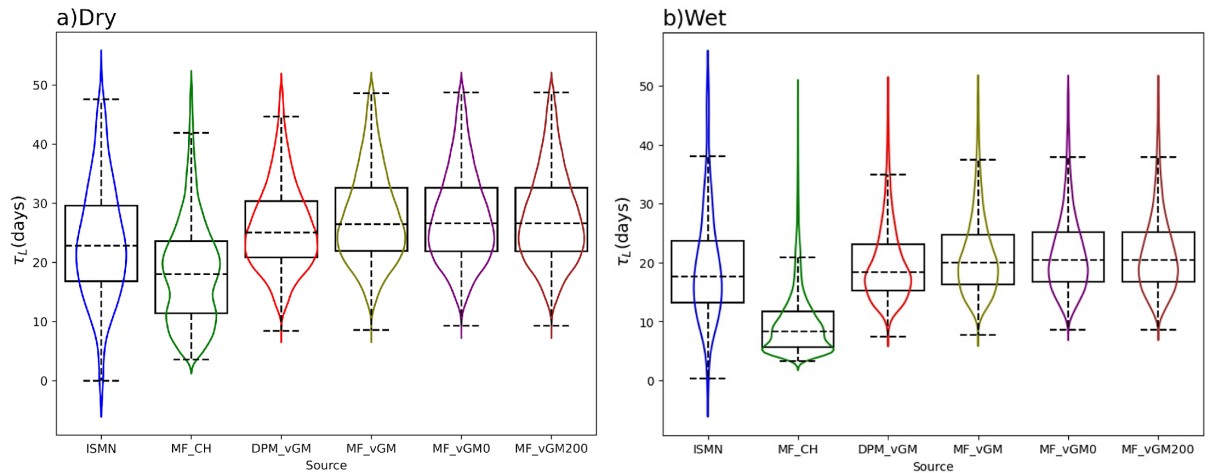

Figure 8 Violin plot of root zone $\tau_L$ estimated from ISMN and Noah-MP scenarios for dry regions with less vegetation (NDVI < 0.45) and wet regions with more vegetation (NDVI > 0.45).

As for the surface layer, we use the MODIS NDVI to classify all the stations into wet and dry regions. In the dry regions (Figure 8a), MF_CH has a different probability distribution function and a lower median of 19 days compared to that of ISMN (median of 23 days). All the other scenarios using VG schemes exhibit a similar SMM PDF to each other, yet they are somewhat different from the one derived from ISMN. Also, the presence of macropores reduces long-term SMM, with a median of 25 days, and results in the closest median to the ISMN (Figure 8a). ISMN, however, shows a large range of long-term SMM compared with all the Noah-MP experiments, indicating the complex nature of the observed SMM needs further investigation (Figure 8a & b). Note that the analyses were conducted at a limited number of locations, presenting challenges in fully capturing the impacts of different parameterizations on SMM.

In the wet regions, MF_CH shows smaller $\tau_L$ values (median of 9.8 days) than that from ISMN (median of 18 days) together with a noticeable pdf difference. The effect of dual permeability decreases the soil moisture and long-term memory compared with the other model experiments, resulting in a median (19 days) close to ISMN (18 days), Figure 8b. However, it seems that the ponding depth does not show a noticeable impact on $\tau_L$. It should be noted that the effect of ponding depth, which slightly increases the long-term memory in RTZ, can be observed in Figure S3 and Figure S4 when we take a close look into them.

Further investigation reveals an enhancement in the model's ability to capture soil hydraulic dynamics when shifting from the Clapp-Hornberger to the Van-Genuchten scheme, with an improvement in $\tau_L$ values from 0.05 to 0.12 (Figure S5). Also, The Dual Permeability model with Van-Genuchten (DPM_VGM) demonstrates superior performance with a correlation of 0.15, compared to all other scenarios tested.

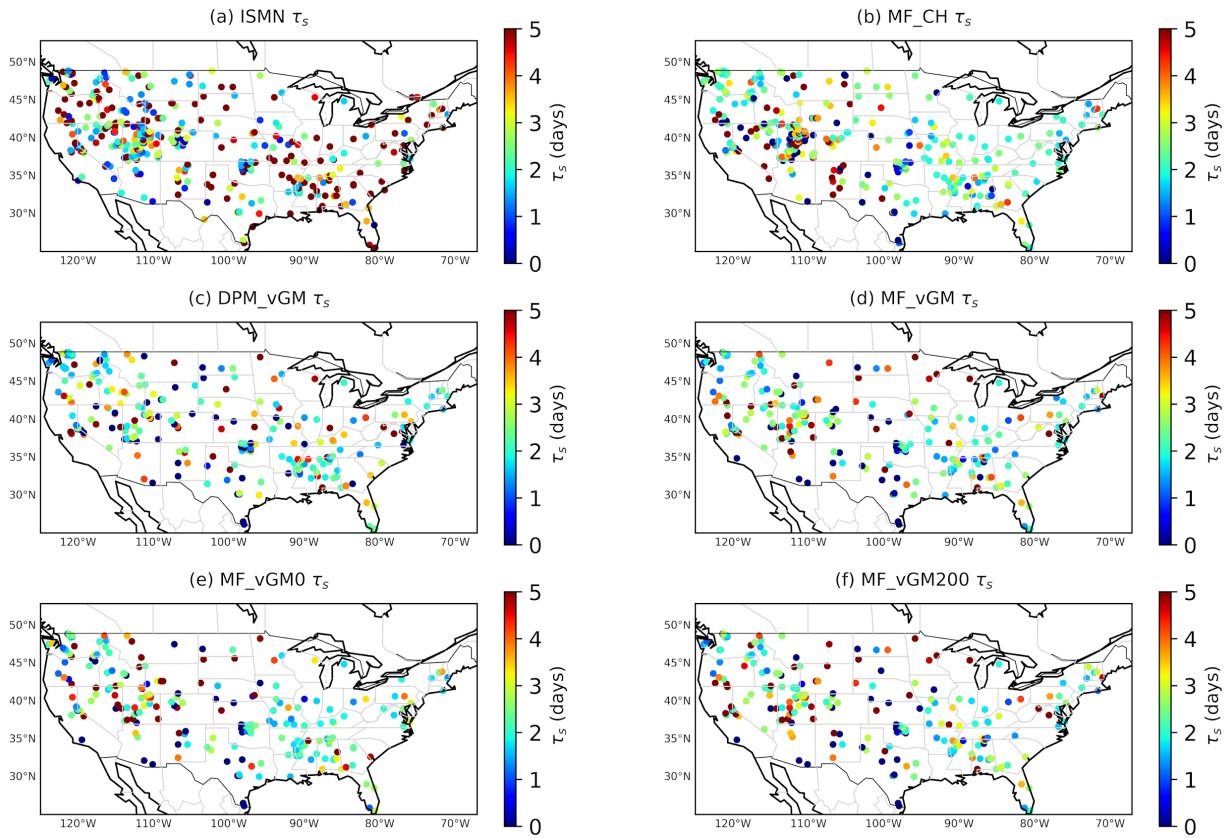

Figure 9 Same as Figure 7 but for short-term.

643

The findings show that $\tau_s$ in most Noah-MP scenarios are comparable to those observed in the ISMN data, as shown in Figure 9b to f. However, there is a consistent underestimation in some eastern locations. Figure 10 highlights this pattern, showing that wet regions tend to underestimate $\tau_s$, with ISMN reporting a median of 2.5 days and Noah-MP experiments a median of around 2 days. Conversely, dry regions tend to overestimate, with ISMN at a median of 2.1 days and Noah-MP experiments at approximately 2.7 days.

Although distinguishing between MF_VGM0 and MF_VGM200 in Figure 9 and Figure 10 is challenging, Figure 11 (Figure 11c and d) reveals that an increase in ponding depth leads to a slight decrease in short-term memory in the eastern CONUS. Comparing Figure 9 with Figure 11 indicates that ISMN stations partially reflect the spatial pattern of long-term and short-term memory in the root zone across CONUS. It may be concluded that the spatial patterns of long-term and short-term memory (Figure 11 and Figure S7) of the root zone are quite similar to those of the surface layer (Figure 3 and Figure 5). Hence, long-term memory is more prevalent in the eastern CONUS and mountainous areas, while longer short-term memory occurs predominantly in western areas. However, this conclusion is not totally true and further investigation is needed.

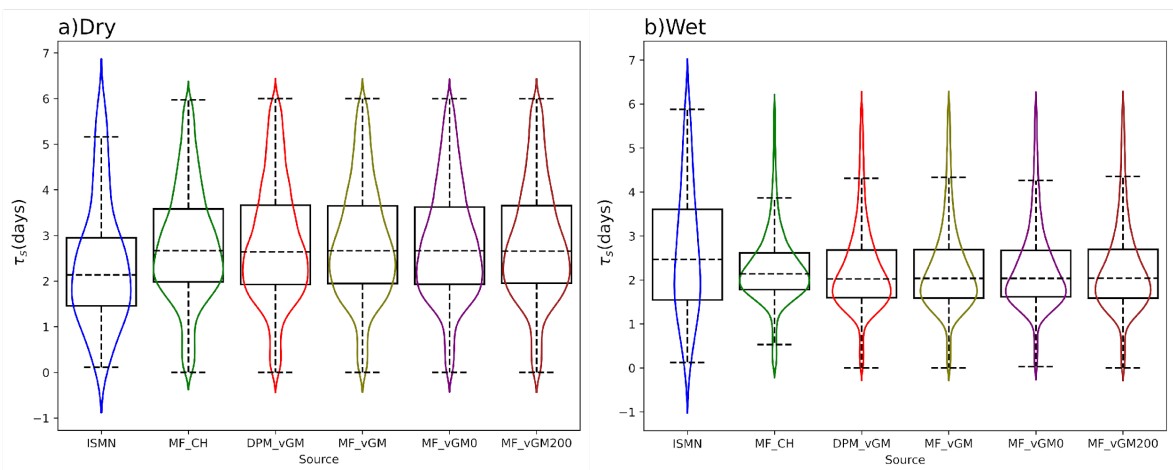

Figure 10 Same as Figure 8 but for the short-term SSM.

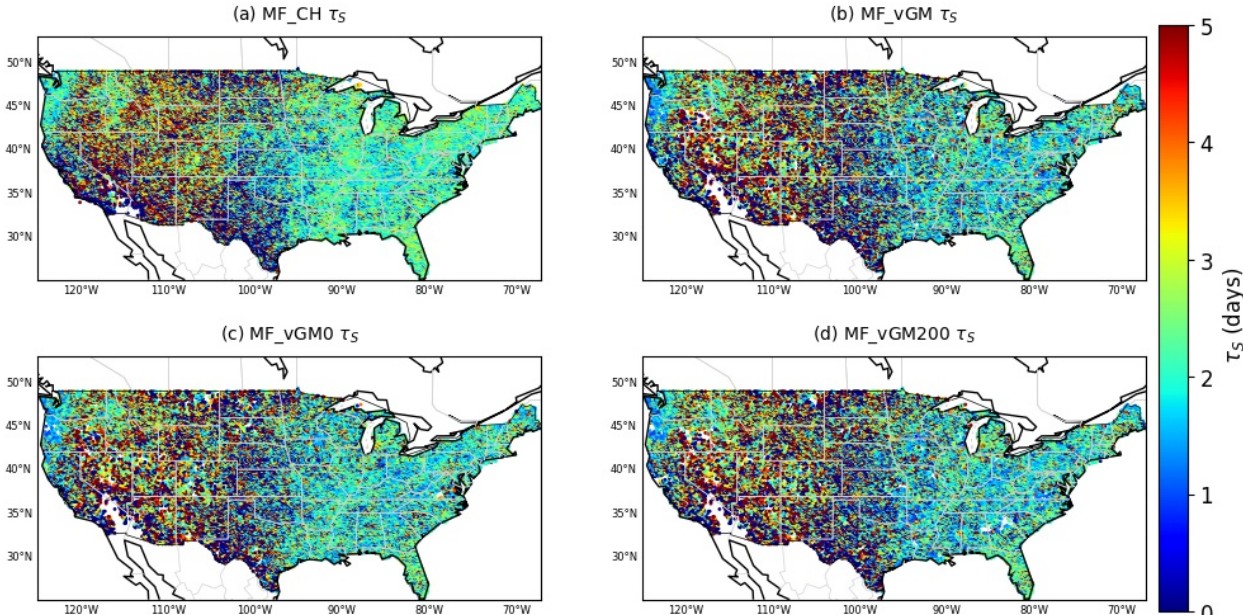

Figure 11 Spatial distribution of root zone $\tau_s$ estimated from (a) MF_CH; (b) MF_VGM; (c) MF_VGM0; and (d) MF_VGM0.

## 4. Discussion

### 4.1 How Do Different Parametrizations Affect SMM?

The efficacy of LSMs in simulating climate feedback mechanisms critically depends on the soil's ability to retain moisture and how fast the soil releases the moisture up to the atmosphere through soil surface evaporation and plant transpiration and down to the aquifers through recharge. The rapid infiltration of incident water (rainfall and snowmelt) into deeper subsoil strata reduces the soil's capacity to return moisture to the atmosphere through evaporation and transpiration. Thereby

disrupting potential atmospheric feedback loops in LSMs (Mccoll et al., 2019). Conversely, If
LSMs lose water too quickly through ET, they provide feedback to the atmosphere faster than they
should. Thus, the concept of SMM becomes essential in LSMs, as it can provide information about
the rate at which moisture disappears from soil. Hence, understanding the effects of various
physical processes on SMM is vital for enhancing the representation of these processes in LSMs,
thereby improving their overall performance in simulating the complex interactions between the
land surface and the atmosphere.
The water retention curve characteristics of the BC/CH hydraulics scheme are characterized by a
strong suction force that is more pronounced than in the Van-Genuchten model for various soil
types (Niu et al, 2024). This stronger suction promotes moisture transfer from the deeper layers to
the surface layer, causing the surface soil to retain more moisture (Figure S6) and has a longer $\tau_L$
(Figure 3, 4), a common issue in LSMs according to He et al. (2023). Moreover, the higher suction
reduces the root zone moisture and consequently, it would have a shorter $\tau_L$ (Figure 7 and 8).
Conversely, the VG scheme, with weaker suction, transfers less moisture from the root zone to the
surface, resulting in a drier surface layer and a shorter $\tau_L$ for the surface, but a longer $\tau_L$ for the root
zone, as depicted in Figures 7 and 8.
Short-term memory is inversely related to moisture availability; thus, a wetter soil has a shorter $\tau_S$,
whereas a drier layer has a longer $\tau_S$. The VG scheme produces a drier surface layer and a moister
root zone, leading to a longer surface $\tau_S$ but a shorter root zone $\tau_S$ compared to the BC/CH scheme,
as shown in Figures 5, 6, and 11.
As indicated in a previous study (He et al., 2023), a common issue in LSMs is the overestimation
of the long-term memory of surface soil over dry regions. This could be attributed to
underestimation of evaporation within LSMs using the CH parametrization (Figure S7a), resulting
in overestimation of soil moisture. However, a shift towards the VG scheme increases the
evaporation (Figure S7b, Figure S8), and hence it overcomes the $\tau_L$ overestimation (Figure 3 and
699   4).

The presence of soil macropores promotes infiltration at the soil surface and rapid flow through
preferential pathways from the surface to the root zone (Mohammed et al., 2021), consequently
reducing the moisture retained in the surface layer. Moreover, macropores lead to reduced suction
of the soil, hence less water from subsurface soil was pulled up to the surface, causing the topsoil
to have less moisture (Figure S6). Therefore, macropores lead to a decrease of surface $\tau_L$(Figure
*3*d, 4b). Moreover, the presence of macropores increases the root-zone soil moisture and
consequently, it should prolong the root zone $\tau_L$. However, the even distribution of macropores
throughout the soil profile in current Noah-MP configuration, DPM_VGM, increases water
infiltration into deeper layers, resulting in faster flow to deep soil layers, recharge to groundwater
and thus a drier root zone. As a result, macropores reduce the root-zone long-term SMM (Figure
7d, e, & f and Figure S8) of DPM_VGM. This highlights the importance of calibration of
macropore profile in DPM_VGM for better representations of macropore effects and soil
hydrohalic dynamics.
While the soil matrix typically allows for only slow water movement due to the pressure gradient,
macropores enable rapid gravitational flow (Mohammed et al., 2018). These macropores facilitate
quicker infiltration to the root zone (Mohammed et al., 2021). Therefore, they increase the drainage
rate to these deeper layers, which slightly reduces the short-term soil moisture memory in the
surface (Figures 5 and 6). Additionally, as water moves from the surface to the root zone, the
increased moisture content there leads to quicker drainage (we speculate that this occurs in the real
world; however, in the current DPM_VGM, the deep soil is wetter than root zone, indicating a
need for calibration of the macropore profile as we have stated). Consequently, this process further
decreases the short-term moisture memory in the root zone due to the higher drainage rates of
wetter soil.

Finally, the ponding threshold allows water to remain on the surface before turning into runoff.
This provides water with more time to percolate into the soil. The consequent increase in ponding
depth allows extended water infiltration, thus enhancing soil moisture and lengthening moisture
retention through the soil profile (Figure S6e, f). So as discussed before, wetter soil leads to
prolonged $\tau_L$ and shorten $\tau_S$ (Figure 5, 6, 7, 11).

## 4.2 Limitation of Our Study


Some sources of uncertainty may affect our results in this study, including uncertainties in input
data, and models. The SMAP L-band penetration depth can indeed be shallower than 5 cm,
especially over wetter regions like the eastern CONUS, which may introduce a mismatch when
comparing SMAP observations with the Noah-MP 5 cm layer. SMAP reliability is affected by
plant water storage change (in the eastern part and some mountainous sites), introducing
uncertainties into SMM values for the benchmark. While SMAP observations may be less reliable
over these densely vegetated areas, they still support our objective of enhancing our understanding
of the physical processes in soil hydrology. Furthermore, the SMM patterns captured from SMAP
can be insightful in understanding regional variabilities in SMM.

Another concern is the influence of ISMN spatial representation on SMM analysis. ISMN stations
are point-based, and it is assumed that one point represents a 1/8-degree grid area. It is possible
that the point measurements cannot fully capture the spatial variability within the Noah-MP grid
cells, leading to discrepancies in the representation of values and spatial patterns. The limited
number of stations may further amplify this issue. One potential solution to address the scale
mismatch between point-based observations and grid-scale simulations is the use of high-
resolution or hyper-resolution models. These models can provide finer spatial detail, allowing for
a more direct comparison between observational data and model outputs, thereby improving the
accuracy of the analysis and reducing scale-induced biases. Incorporating such approaches in
future studies would help mitigate the limitations posed by the current scale differences.

Additionally, some model representations may require further investigation. The DPM_VGM
scheme uses vertically constant macropore volume fraction, which means macropores generated
by biotic factors (formed by wormhole and dead roots) and abiotic factors (cycles of freezing-
thawing and drying-wetting) are fixed down to the bedrock. However, in nature, these macropores
would reduce after a few meters from the soil surface. Because the existence of macropores in
nature drains the surface layer and increases the root zone soil moisture, to better represent the
actual physical process, it is necessary to incorporate more soil data, e.g., the soil organic matter
and coarse materials from e.g., SoilGrid250m  for climate predictions or calibrate macropore
volume fraction for hydrological applications. Such a calibration is anticipated to further advance
the fidelity of soil moisture simulations, enhancing the model's utility in various hydrological and
climate applications.
Concerning surface water ponding, a constant ponding threshold may not be justified, and a
spatially variable surface ponding may lead to improved model accuracy. Future model
developments should consider micro-scale topographic variations to represent the hydrologic
connectivity of surface ponded water. We tested a scheme of ponding threshold as a linear function
of the subgrid standard deviation of DEM derived from DEM at 30 m resolution (not enough
though), resulting larger surface ponding thresholds over the alpine west US. Further investigation
is needed to validate and calibrate the modeled areal ponding fraction and depth against satellite
(or camera) derived. We expect a more realistic representation of ponding threshold through
further calibration of the parameters in the function.
There are additional factors, such as water convergence through surface and subsurface lateral
flows, that may affect SMM but are not represented by the current Noah-MP version and thus not
considered in our analysis. The primary focus of our study is to understand the impacts of missing
processes on SMM and use this understanding to guide future LSM development for S2S climate
predictions, for instance, the surface ponding and preferential flow. Consequently, we narrowed
our examination down to key missing processes represented within Noah-MP. Future research
would further evaluate the impact of lateral flows and other processes on SMM, expanding our
understanding of these dynamics and their implications for climate prediction. Moreover, this
study focuses primarily on physical process representations and parameterizations for soil moisture
dynamics, while we acknowledge the strong impacts of uncertainties in hydraulic parameters.
**5. Conclusion**
In this study, we have explored the effects of soil hydraulic schemes and hydrological processes
on SMM using the Noah-MP LSM with advanced hydrology. Our research was motivated to
understand how missing physical processes help solve the commonly observed biases in long-
term/short-term SMM by LSMs. We aim to find the key missing processes controlling SMM and
thus to improve the representation of soil hydrology in LSMs, using the knowledge gained from
our analysis of SMM. We designed and implemented five scenarios to focus on the impacts of key
missing processes and different hydraulic parametrizations. These scenarios include two soil
hydraulic models (Clapp and Hornberger and Van-Genuchten), a dual permeability model
representing preferential flow, and three surface ponding thresholds. Using soil moisture datasets
from SMAP and ISMN for surface and root zone measurements, respectively, we conducted a
comprehensive analysis of the effects of different Noah-MP parameterizations on soil moisture
memory.
Our findings suggest that the soil water retention curve is the most important factor controlling
SMM, due to its strong influence on soil water persistence through suction by the soil particles.
We show that the adoption of the Van-Genuchten (VG) parameterization considerably mitigates
the long-standing issue of overestimating SMM in LSMs employing the Brooks-Corey/Clapp-
Hornberger (BC/CH) hydraulic model. The Van-Genuchten model, with its reduced suction effect
attributable to a drier surface layer, leads to a more accurate depiction of moisture transfer from
the root zone to the surface, which is important for more realistic description of soil moisture
dynamics.
Moreover, representing surface ponding processes allows for an extended period of soil water
infiltration, thus extending both surface and root-zone long-term memories and reducing the short-
term memory. Implementing a dual-permeability approach fine-tunes soil moisture representation
by accounting for preferential flow paths, marking a step forward in the enhancement of soil
moisture memory and the overall fidelity of hydrological simulations. Macropores lead to a
decrease in short-term memory and long-term memory, due to faster drainage and thus decreased
surface soil moisture. Given these compelling advancements, we strongly recommend that LSMs
adopt the VG hydraulics to advance the prediction of hydrological and climatic phenomena.
The findings from this study have important implications for future research on SMM. By
identifying the specific parameterizations that lead to discrepancies in long-term and short-term
SMM, future studies should focus on refining these parameters to reduce biases in LSMs.
Moreover, while this study focuses on the effect of the missing hydrological processes on the
timescale of SMM, future research should analyze the impact of these parameterizations on the
strength and legacy of SMM and assess whether the findings based on timescale align with those
related to strength and legacy (Rahmati et al., 2024).

**Competing interests**
The contact author has declared that none of the authors has any competing interests.
**Acknowledgments**

Funding for this project was provided by the National Oceanic and Atmospheric Administration (NOAA), awarded to
the Cooperative Institute for Research on Hydrology (CIROH) through the NOAA Cooperative Agreement with The
University of Alabama, NA22NWS4320003. Also, the research carried out for this article was supported by the U.S.
Army Corps of Engineers, Engineer Research and Development Center, Coastal Inlets Research Program via
Congressionally Directed R&D with the National Oceanic and Atmospheric Administration's National Water Center.
The data used in this study are freely available online:
NLDAS-2 data (http://www.emc.ncep.noaa.gov/mmb/nldas/); NASA SMAP soil moisture product
(https://nsidc.org/data/spl3smp_e/versions/6); GPM IMERG-Final product
(https://disc.gsfc.nasa.gov/datasets/GPM_3IMERGHH_06/summary). The Noah-MP code used in this study has
been uploaded to a repository that may be accessed by other researchers
(https://github.com/mfarmani95/NoahMP_Dual).

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
