# Peer review of "Plain Language Summary"

_EGUsphere, 2024_

## Author Comment (AC2)

**The First Reviewer**

It's my pleasure to review this manuscript on NoahMP model. The study utilized the Noah-MP land surface model, implementing five different soil hydraulic parameterization schemes, including two soil hydraulic models, a dual-permeability infiltration scheme, and variations in surface ponding depth. By integrating soil moisture datasets from SMAP and ISMN, the study explored the effects of these schemes on Soil Moisture Memory (SMM). It was found that using the Van-Genuchten parameterization and dual-permeability infiltration scheme improved the simulation of SMM. Considering surface ponding extended soil moisture infiltration, thereby improving moisture conditions in the surface and root zone, leading to increased long-term memory and decreased short-term memory. Conversely, using macropores reduced SMM. In general, the manuscript is well written and fully within the scope of HESS. While I still have some concerns before the recommendation of its publication.

**Major:**

1. When designing the five different Noah-MP model scenarios, how did the authors determine the specific parameter values (such as the ponding depth threshold)? Were these parameter values selected based on previous literature, sensitivity analyses, or other considerations?

   R: In our study, we conducted sensitivity analyses for key parameters such as the ponding depth threshold, macropore fraction, and other relevant factors. The initial parameter values were determined based on testing various configurations to assess their influence on the model's performance in modeling surface soil moisture, runoff peaks, and baseflow. These tests allowed us to identify a range of values that produced realistic outcomes, but we acknowledge that the selected values are not final. Further parameterization scheme and calibration is necessary to optimize these parameters, and this will be part of our future work to refine the model's accuracy.

   The following is added to paper: "While we conducted sensitivity analyses on key parameters such as the ponding depth threshold and macropore fraction to identify ranges yielding realistic outcomes, we acknowledge that further calibration is necessary to optimize these parameters and will address this in future work to enhance the model's accuracy."

2. In Figure 4, the four VGM schemes exhibit greater variability in wet regions, unlike Figure 6 where their PDF distributions are generally consistent in both arid and wet regions. Ji et al. (2023) enhanced the synergistic effect of high-resolution soil dataset and model parameterizations. Here, this discrepancy may be also caused by differences in how each VGM scheme incorporates local soil moisture dynamics and interactions with vegetation, leading to varying responses across different hydrological regimes.

Reference: Ji, P., Yuan, X., & Jiao, Y. (2023). Synergistic effects of high-resolution factors for improving soil moisture simulations over China. Water Resources Research, 59, e2023WR035513.

R: The following is added to paper:
"The modeling results also indicate the long-term memory of the surface soil moisture is more sensitive to the four VGM schemes in the wet regions (Figure 4b) than the short-term memory (

Figure 6b). This can be attributed to the differences in how topsoil water responds to surface ponding and preferential flow as represented by the four VGM across different moisture regimes. Under higher soil moisture conditions right after a rainfall event, the persistence of soil moisture is mainly dominated by drainage of topsoil water to deeper soil, whereas at relatively lower soi moisture, the long-term memory is more controlled by persistent water inputs from surface ponded water and prolonged drainage by preferential flow. This also indicates that the effects infiltration of surface ponded water and preferential flow can last longer up to more than 10 days. Under dry conditions (Figure 4a and 6a), these hydrological processes become less important. However, the soil water retention curves as represented by the CH and VG schemes play a more important role under any conditions (Figure 4a and Figure 6a). Another possible reason could be the issue of time scale. Short-term memory has values up to 5 days, and given the SMAP revisit time of 3 days, generating values for intervals shorter than 3 days may challenge the validity of short-term memory as a reliable measurement for soil drainage, as demonstrated by McColl et al. (2019). Since we selected Noah-MP days corresponding to the SMAP revisit time, it is possible that the effects of different VG parameterizations were diminished by this sampling. We suggest that other measurements, such as streamflow and baseflow analysis, should be considered to better quantify the effect of soil hydraulics on soil drainage."

When comparing different model scenarios with SMAP and ISMN data, the study primarily focuses on the median of SMM. However, the shape and range of SMM distributions could also provide valuable information. Analyzing discrepancies between model-generated SMM distributions and observational data can offer a more comprehensive assessment of model performance.

R: We acknowledge that analyzing the shape and range of Soil Moisture Memory (SMM) distributions could provide valuable insights into the model's performance relative to observational data. However, the primary focus of this study is to assess the general trends and central tendencies of SMM by comparing the median values across different model scenarios. The use of median SMM serves as a robust metric that reduces the influence of outliers and extreme values, offering a clearer comparison of core model performance.

Hence the following is added to the paper: "While analyzing the shape and range of Soil Moisture Memory (SMM) distributions could offer valuable insights, we focus on comparing median SMM values across model scenarios to assess general trends and central tendencies, thereby reducing the influence of outliers for a clearer evaluation of model performance."

**Minor**
1. Lines 399-447, The article extensively describes various parameterization schemes of the Noah-MP model, but could further discuss the underlying physical mechanisms of these schemes to help readers better understand their impact on SMM.

   R: Yes, we totally agree because this paper focuses on "why" not "accuracy." There is another paper in which we provide more details of the underlying physics, hence the following sentences is added to the paper: "A detailed description the underlying physical mechanisms of the schemes used in this

study could be found at Niu et al, (2024), also a brief description of equations and parameters is included in supporting material" We also revised the paper, in the result analysis, to add the underlying mechanisms to explain the modeling results, although we already provided explanations in the Discussion section.

2. Lines 466-467"The MF_CH experiment displays a spatial pattern that contrasts with the SMAP data, with a longer memory in the arid western regions but a shorter memory in the wet northeastern regions " What could be the reasons for this spatial distribution?

   R: We added an explanation to the paper for it:
   " This is likely caused by the faster drainage of topsoil water under the wetter conditions, whereas under the drier conditions, the spuriously stronger suction from the CH hydraulics sustain the surface soil moisture for a longer period."

3. While discussing the limitations of ISMN data, the paper mentions the issue of scale differences between point measurements and grid-scale data. I suggested to added some discussions on the high-resolution or hyper-resolution, which might be an efficiency way to solve the scale mismatch between observation and simulation.

   R:        We        added        a        paragraph        updated        to:
   "Another concern is the influence of ISMN spatial representation on SMM analysis. ISMN stations are point-based, and it is assumed that one point represents a 1/8-degree grid area. It is possible that the point measurements cannot fully capture the spatial variability within the Noah-MP grid cells, leading to discrepancies in the representation of values and spatial patterns. The limited number of stations may further amplify this issue. One potential solution to address the scale mismatch between point-based observations and grid-scale simulations is the use of high-resolution or hyper-resolution models. These models can provide finer spatial detail, allowing for a more direct comparison between observational data and model outputs, thereby improving the accuracy of the analysis and reducing scale-induced biases. Incorporating such approaches in future studies could help mitigate the limitations posed by the current scale differences."

4. Lines 712-713:" ... processes that influence SMM and to address the commonly observed overestimation/underestimation of long-term/short-term SMM in LSMs." I suggest to add some implications of these findings for future research on SMM.

R: Based on the Rahmati et al (2024) suggestions we added the following to the paper: "The findings from this study have important implications for future research on SMM. By identifying the specific parameterizations that lead to discrepancies in long-term and short-term SMM, future studies can focus on refining these parameters to reduce biases in LSMs. Moreover, while this study focuses on the effect of different parameterizations on the timescale of SMM, future research can analyze the impact of these parameterizations on the strength and legacy of SMM and assess whether the findings based on timescale align with those related to strength and legacy."

**Citation**: Rahmati, M., Amelung, W., Brogi, C., Dari, J., Flammini, A., Bogena, H., Brocca, L., Chen, H., Groh, J., Koster, R. D., McColl, K. A., Montzka, C., Moradi, S., Rahi, A., Sharghi S., F., and Vereecken, H.: Soil Moisture Memory: State-Of-The-Art and the Way Forward, Reviews of Geophysics, 62, e2023RG000828, 2024.

---

## Author Comment (AC3)

**The Second Reviewer**
Soil moisture memory (SMM) is a key characteristic describing land-atmosphere interactions. The authors used the latest version of NOAH-MP with several parameterization schemes to simulate the SMM and compared it with the SMM derived from SMAP and ISMN observations. The manuscript is well organized and written. I have only a few minor concerns as follows.

1. The authors used the SOM modeled by NOAH-MP to estimate the Marcropore Volume Fraction (MVF). What about the reality and accuracy of these SOM maps? And then, how is the error in the SOM estimate transferred to the MVF? And finally, what is the impact of uncertainty on the MVF results?

   R: A good questions. Instead of using a spatially-constant MVF, we tried to link MVF with SOM to use spatially-varying MVF. However, this representation of the MVF in the real world apparently not enough, because it reflects only the biotic factors not other abiotic factors, e.g., MVF formed during freezing-thawing cycles (especially in the seasonal frozen ground and thawing permafrost regions) and cracks formed during wetting-drying clays. To represent these processes is even more challenging. We have also tested machine-learning based, 1 km, global datasets of hydraulic parameters (should contain macropore information) but it does not show significant differences from other macropore simulations.

   Introducing MVF reduces the numbers of parameters that need to be calibrated. In practice, we can calibrate MVF, one parameter, instead of Ksat and other water retention curve parameters. Future studies should learn this important parameter from soil moisture profile or streamflow, e.g., using differential modeling.

   From the indications as stated in the paper, a larger value of MVF can substantially enhance drainage of soil water (while reducing the water retention) to deeper soils and aquifers (increase recharge) and thus reducing both short-term and long-term memories.

2. For the SMM observed by the ISMN stations, the authors show the results of the root zone layers. What about the results in the surface layer? It is better to show the SMAP, NOAH-MP, and ISMN results at the same time.

   R: We acknowledge the suggestion to include surface layer results for ISMN stations alongside SMAP and Noah-MP data. However, there are two key reasons why ISMN data was not used for the surface layer in our analysis. First, the spatial scale of ISMN observations is at the point level, which is significantly smaller than the 1/8-degree grid scale of Noah-MP and SMAP, making direct comparisons challenging due to potential scale mismatch. Additionally, the limited number of ISMN stations across the U.S. further complicates the ability to represent spatial patterns at a regional or national scale.

   In contrast, SMAP provides continuous surface soil moisture data across the entire U.S. with a resolution that is much closer to that of Noah-MP, allowing for a more consistent comparison. Furthermore, SMAP has been widely used in previous studies as a benchmark

for model validation, enabling us to compare our results with those established in the literature. For these reasons, we focused on SMAP and Noah-MP for surface layer analysis while using ISMN data primarily for root zone validation, where its limitations are less critical.

The following part in the paper is modified to reflect your concern: "Some sources of uncertainty may affect our results in this study, including uncertainties in input data, and models. The SMAP L-band penetration depth can indeed be shallower than 5 cm, especially over wetter regions like the eastern CONUS, which may introduce a mismatch when comparing SMAP observations with the Noah-MP 5 cm layer. SMAP reliability is affected by plant water storage change (in the eastern part and some mountainous sites), introducing uncertainties into SMM values for the benchmark. While SMAP observations may be less reliable over these densely vegetated areas, they still support our objective of enhancing our understanding of the physical processes in soil hydrology. Furthermore, the SMM patterns captured from SMAP can be insightful in understanding regional variabilities in SMM.

Another concern is the influence of ISMN spatial representation on SMM analysis. ISMN stations are point-based, and it is assumed that one point represents a 1/8-degree grid area. It is possible that the point measurements cannot fully capture the spatial variability within the Noah-MP grid cells, leading to discrepancies in the representation of values and spatial patterns. The limited number of stations may further amplify this issue. One potential solution to address the scale mismatch between point-based observations and grid-scale simulations is the use of high-resolution or hyper-resolution models. These models can provide finer spatial detail, allowing for a more direct comparison between observational data and model outputs, thereby improving the accuracy of the analysis and reducing scale-induced biases. Incorporating such approaches in future studies would help mitigate the limitations posed by the current scale differences."

3. What physical situation in reality corresponds to a water pond of several tens to two hundred millimeters thick on a grid of 0.125 degrees in the model? How long does it last? Is it physically realistic?

R: Surface ponded water results from the exceedance of precipitation rate over infiltration rate. It occurs over regions where the soil's permeability is low (clay soil and frozen ground) during severe storms or spring snowmelt. About a quarter of the severe floods in the US in the 20th century were directly linked to surface ponded water from spring snowmelt.

The ponding threshold highly depends on the hydrological connectivity of surface ponded water in local depressions that are further dependent on micro-scale topography. In the beginning when we design this model, we do not have any idea about its value or range. After testing against streamflow observations, we know this value can be up to ~100 mm.

[Figure]

200 mm is to make sure there is no runoff of surface ponded water (infiltration excess runoff); 0 mm means all excessive water runs off.

4.  In the caption of Figure 1, the last sentence, Theta_w is missing.
    R: Paper is revised

**Citation**: https://doi.org/10.5194/egusphere-2024-1256-RC2

---

## Author Comment (AC4)

The Third Reviewer

It's my pleasure to review egusphere-2024-1256 "What Are the Key Soil Hydrological Processes to Control Soil Moisture Memory?" by Farmani et al. The authors conducted several numerical experiments using the Noah-MP LSM to investigate the impact of soil water retention characteristics (or soil hydraulics), soil permeability (or preferential flow), and surface ponding on simulating profile soil moisture dynamics as well as capturing the long-term and short-term soil moisture memory as observed by SMAP satellite and in situ ISMN networks over the contiguous United States (CONUS). The research is very interesting and should be useful for improving the simulations of soil moisture dynamics and memory using the LSM. However, the design and validation of numerical experiments can be further improved given the fact that several other processes (e.g. infiltration, evapotranspiration and drainage) affecting the simulation of soil water flow are not well considered. In addition, detail descriptions on the adopted Noah-MP model should be provided since the version adopted in this paper is not an official released one. Accordingly, major revision is recommended. My comments are as follows.

Thank you for reading the paper and your valuable feedback!

**Major:**
1. The design of numerical experiments should be improved to consider the impact of several other processes (e.g. infiltration, evapotranspiration and drainage) and hydraulic parameters on simulating profile soil moisture dynamics and memory.

R: Thanks for the comments, which force us to rethink the main purpose of this this study.

The focus of this paper is primarily on the impacts of key hydrological processes that may be missed by mainstream LSMs on SSM, e.g., preferential flow and surface ponding. So we changed the title to
"Do Land Models Miss Key Soil Hydrological Processes Controlling Soil Moisture Memory?" instead of
"What are the Key Soil Hydrological Processes Controlling Soil Moisture Memory?"

Accordingly, we revised the Abstract, Introduction, and Discussion, and Conclusion sections. Please see the doc with tracked changes.

Actually, in the review paper of Rahmati et al. (2024), they have summarized the major mechanisms of soil moisture memory emergence, including all factors affecting ET and drainage (10s of papers on the effects). Experiments to investigate the effects and ET and drainage would make this paper too general.

2. Besides the physical process and parameterization, the uncertainty related to hydraulic parameters could also affect the simulation of soil moisture dynamics. The authors should at least discuss the potential impact of this.

R: Thank you for your insightful comment. While we acknowledge that uncertainty in hydraulic parameters can impact soil moisture dynamics, our prior objective of this research is to investigate the missing physical processes (model structures) instead of uncertainties in model parameters. Although uncertainties in hydraulic parameters may reflect uncertainties in model structure and finally can solve the problem e.g., directly increase the hydraulic conductivity (to mimic the preferential flow model), but it does not help understanding the effects of biotic (worm holes and dead roots etc.) and abiotic factors (freezing-thawing cycles and drying-wetting cycles).

We have revised the Discussion Section, accordingly.

**Minor:**
1. The validation of numerical experiments can be improved given the fact that the ISMN networks also provide measurements of surface soil moisture. The authors can consider validating the model simulations at point/pixel scale using soil moisture measurements at both surface and deeper layers from the ISMN networks, and using the SMAP product for regional-scale validation. In addition, the penetration depth of L-band can be shallower than 5 cm, how the mismatch between the sampling of SMAP satellite and model layer thickness affect the validation?

   R: The primary aim of this study is to analyze the effects of missing physical processes on soil moisture dynamics rather than to conduct a full validation of the Noah-MP model. While ISMN data does provide valuable point-scale surface soil moisture measurements, the spatial scale of these observations is considerably smaller than the 1/8-degree grid scale of Noah-MP and SMAP. This scale mismatch makes direct comparisons challenging for surface soil moisture, especially given the spatial heterogeneity at point scales.

   SMAP, on the other hand, provides continuous surface soil moisture data across the U.S. at a spatial resolution closer to that of Noah-MP, allowing for a more consistent comparison. SMAP is also widely recognized in the literature as a benchmark for model validation, enabling us to align our results with those from previous studies. For these reasons, we focused on SMAP for surface layer analysis and used ISMN data primarily for root zone validation, where point-level observations are less impacted by scale limitations.

   The SMAP L-band penetration depth can indeed be shallower than 5 cm, especially over wetter regions like the eastern CONUS, which may introduce a mismatch when comparing SMAP observations with the Noah-MP 5 cm layer. SMAP's reliability is particularly affected by plant water storage changes in the eastern U.S. and specific mountainous areas, which can introduce variability in SMM values when used as a benchmark. While SMAP observations may be less reliable over these densely vegetated areas, they still support our objective of enhancing our understanding of the physical processes in soil hydrology. Additionally, the SMM patterns observed from SMAP offer valuable insights into regional variability, which aligns with our study's goals.

   The following part in the paper is modified to reflect your concern: "Some sources of uncertainty may affect our results in this study, including uncertainties

in input data, and models. The SMAP L-band penetration depth can indeed be shallower than 5 cm, especially over wetter regions like the eastern CONUS, which may introduce a mismatch when comparing SMAP observations with the Noah-MP 5 cm layer. SMAP reliability is affected by plant water storage change (in the eastern part and some mountainous sites), introducing uncertainties into SMM values for the benchmark. While SMAP observations may be less reliable over these densely vegetated areas, they still support our objective of enhancing our understanding of the physical processes in soil hydrology. Furthermore, the SMM patterns captured from SMAP can be insightful in understanding regional variabilities in SMM.

Another concern is the influence of ISMN spatial representation on SMM analysis. ISMN stations are point-based, and it is assumed that one point represents a 1/8-degree grid area. It is possible that the point measurements cannot fully capture the spatial variability within the Noah-MP grid cells, leading to discrepancies in the representation of values and spatial patterns. The limited number of stations may further amplify this issue. One potential solution to address the scale mismatch between point-based observations and grid-scale simulations is the use of high-resolution or hyper-resolution models. These models can provide finer spatial detail, allowing for a more direct comparison between observational data and model outputs, thereby improving the accuracy of the analysis and reducing scale-induced biases. Incorporating such approaches in future studies would help mitigate the limitations posed by the current scale differences."

2. Detail descriptions on the adopted Noah-MP model should be provided since the version adopted in this paper is not an official released one. For instance, how many options are available for each process listed in Tables 1 and 2, and what are the criteria for determining the options listed in Table 1? Particularly, detailed introduction (e.g. relevant equations) of the processes and options related to the simulations of soil moisture dynamics is necessary, which can be included in the Appendix.

R: Thank you for the suggestion to provide additional details on the Noah-MP model version used in this study. The specifics of the underlying physics, parameters, and details related to this new version of Noah-MP are thoroughly documented in a separate paper, where we cover all parameters listed in Table 2. Additionally, the options presented in Table 1 are derived from the existing publicly available Noah-MP model, and all options are described in detail in Niu et al. (2011) and later in He et al. (2023), which is publically accessible. Together, these references provide comprehensive coverage of the parameters, equations, and process options related to soil moisture dynamics in our simulations.

We summarize the equations of Niu et al (2024), which is included in the Supporting Information. Also, we provide the updated model code through a github link: (https://github.com/mfarmani95/NoahMP_Dual).

He, C., Valayamkunnath, P., Barlage, M., Chen, F., Gochis, D., Cabell, R., ... & Ek, M. (2023). *The community Noah-MP land surface modeling system technical description version 5.0* (p. 5). NCAR Technical Note NCAR/TN-575+ STR, doi: 10.5065/ew8g-yr95.

3. L213: I suggest to introduce the model/experimental design and datasets in two separate sections.

   R: The manuscripts is revised accordingly

4. L236 and others: the number of sections and subsections in the manuscript is wrong.

   R: The manuscripts is revised accordingly

5. L288-292: it's not clear how the authors use the IMERG precipitation product to run the model?

   R: To incorporate the IMERG precipitation product, we modified the forcing component of the Noah-MP code. Specifically, we used an average of NLDAS-2 and IMERG precipitation in cases where NLDAS-2 generated negative precipitation values, which was particularly relevant for coastal regions. This adjustment helped improve the accuracy of precipitation inputs in these areas.
   The following is added to the paper:
   "To integrate the IMERG precipitation product into the model, we modified the forcing component of the Noah-MP code. Specifically, an average of NLDAS-2 and IMERG precipitation was employed when NLDAS-2 reported negative precipitation values, which was particularly significant in coastal regions. This adjustment enhanced the accuracy of precipitation inputs, contributing to more reliable simulations in these areas."

6. L307-300: the time span is 2015-2019 as given in Abstract, please clarify. In addition, why the authors only choose the product of these years give the fact that the product is available up to now.

   R: We selected the 2015-2019 time span to ensure consistency with previous studies, specifically MacColl et al. (2019) and He et al. (2023).

7. L404-405: did the groundwater level reach equilibrium after only five iterations (i.e. 25 years for spin-up)?

   R: a good question! For the groundwater level to reach an equilibrium state, it would take more than 100 years and even longer e.g., 1,000 years in some regions, e.g., the drylands of Arizona with a thick vadose zone (100 meters) with very slow processes like vaporization and diffusion of groundwater to the ground. However, as the critical zone affecting surface processes and the top 2 m soil moisture dynamics is ~5 meters (Kollet and Maxwell, 2008), indicating that the water table below ~5 meters would have negligible effects.

For the top 2m soil moisture to reach an equilibrium, it takes several years with a maximum of 10 years in most cases. This is also consistent with the soil moisture memory analysis. It is generally accepted that the soil moisture memory is important for climate predictions at S2S scales (the ocean's memory lasts longer).

Kollet, S. J., and R. M. Maxwell (2008), Capturing the influence of groundwater dynamics on land surface processes using an integrated, distributed watershed model, Water Resour. Res., 44, W02402, doi:10.1029/ 2007WR006004.

8. L406-409: I think detail descriptions should be provided, e.g. provide a list of parameter values adopted in this study.

R: There is another paper in which we provide the detail of the underlying physics, parameters and all the details related to this new version of Noah-MP.

Niu et al, (2024) is publicly available online:
https://d197for5662m48.cloudfront.net/documents/publicationstatus/216663/preprint_pdf/7f3141f24161bbc7b24e562e983640b2.pdf

Because it is published, we provide more details of the model descriptions and parameters in the SI.

9. L416-421: the official version of Noah-MP only include four soil layers with depth up to 2m, how the authors solve both soil water and heat flow processes with the new configuration of layer thickness and depth?

R: There is another paper in which we provide the detail of soil moisture solver, hence the following sentences is added to the paper:
"A detailed description the underlying physical mechanisms of the schemes used in this study could be found at Niu et al, (2024), also a brief description of equations and parameters is included in supporting material"

10. L448: for the results presented, how the authors match the model simulations with SMAP product? For instance, the SMAP product include ascending and descending overpasses, how the authors match their simulations results with these?

R: In the paper we mentioned "In this study, we selected the SMAP Level 3 morning overpass due to the greater likelihood of air and surface temperature equilibrium during these hours, a critical condition for the SMAP retrieval algorithm."

---

## Author Comment (AC5)

**CC1:** 'Comment on "What Are the Key Soil Hydrological Processes to Control Soil Moisture Memory?" by Farmani et al. (2024)'**, by Mehdi Rahmati, 23 May 2024**

It was with great interest that I read this interesting article written by Farmani et al. (2024). After reviewing and reading almost all works on soil moisture memory (SMM), it must unfortunately be noted that the effects of soil properties on SMM are very rarely investigated, and it's great to see that a research group has conducted such interesting research directly on this topic. To emphasize the importance of this and all similar research looking at SMM and the link with soil properties, I may copy and paste here the part of the "The way forward" section of our review paper on SMM, which has just been published in Reviews of Geophysics (Rahmati et al. (2024); https://agupubs.onlinelibrary.wiley.com/doi/10.1029/2023RG000828):

"Finally, SMM is the result of a complex interplay of physical, biological, and hydrological processes and soil properties (Group 3) (Rahmati et al., 2023). In fact, SMM is rooted in the integrative nature of soil moisture as a water reservoir (Orth and Seneviratne, 2013), which can be influenced by multiple processes (Figure 3), including soil infiltration, soil water redistribution and storage, root water uptake, capillary rise, and drainage. This review shows that the literature, in general, considers soil depth and soil porosity (as it appears in the autocorrelation expression) to be the main soil properties controlling SMM. While we recognize the valuable contributions of previous efforts such as the SoilWat initiatives (e.g., Aliku and Oshunsanya, 2018; Andrews and Bradford, 2016; Oyeogbe and Oluwasemire, 2013), we maintain that additional consideration should be given to pore size distribution, soil mineral composition (e.g., type and amount of clay), soil organic carbon, and other such properties, as these can control water retention, hydraulic conductivity, and diffusivity and accordingly can influence SMM. In addition, the importance of "hydraulic redistribution" by roots (Dawson, 1993), which is of prominent importance during dry periods by bringing water from deep reservoirs to the near surface soil (Caldwell et al., 1998; Jackson et al., 2000), needs to be emphasized in future research. Hagemann and Stacke (2015) have already shown that hydraulic redistribution by a wide range of plant species is significant in many different biomes around the globe and has implications for SMM."

The improper integration of soil memory (as a comprehensive concept that includes SMM, as soil moisture is only one of the carriers of memory in the soil) into LSMs has already been highlighted in another article of our group published in Nature Reviews Earth & Environment (Rahmati et al. (2023); https://www.nature.com/articles/s43017-023-00454-5) where we have already stated that LSMs neglect soil memory.

Finally, although I do not put myself in the shoes of the reviewers of this paper and therefore leave the technical comments to them, I have only one concern that I thought might be overlooked and would be better to mention, namely that in lines 164 to 174, you mentioned several metrics for quantifying SMM (of course you can find more, as listed in our review) and then stated in lines 175 to 177 that "These methods provide insights into the magnitudes of water and energy flux exchanges between land surface and atmosphere, indicating that shorter SMM durations can lead to more intense feedback and larger flux exchanges". However, I think that such an insight cannot be adopted so easily. After reviewing almost all available work on SMM, I can only say that the SMM timescale merely indicates the duration or time window within which the current state of the soil moisture causes feedback to the land surface process. However, we cannot judge from the

 As we mentioned in our review, future research on SMM should examine the strength of SMM in addition to its timescale. So far, the only criterion to investigate the strength of the feedback is the autocorrelation value itself. All the other criteria you have listed here only quantify the SMM timescale, of course, mostly based on the autocorrelation, **which certainly cannot say anything about the strength of the feedback**. In short, the SMM timescale only defines the active period of memory (see the following figure, which is copied from Figure 1 of our review paper), **not its strength**. I may be wrong, but this is how I can understand it, even from a mathematical point of view.

[Figure]

To summarize, I would say that an extreme event (exogenous or endogenous, whatever it is) leads to Soil Memory (as a whole, which includes SMM), which is only a descriptive phenomenon to describe the process as a whole: a phenomenon that occurs in the soil (or we can call it an emergent property of the soil, so emerges in soil) that describes how and why information is fed into the soil after a single event or series of events, how the information is stored, and transferred across the time axis, and what mechanisms are involved and how they affect the variables, fluxes, and functioning of the future system. However, when it comes to quantifying it, we can assume three different characteristics, including timescale, strength, and legacy effects:

Soil Memory Timescale: the time period in which the soil can remember these effects. If the carrier is known (e.g., soil moisture, soil carbon, etc.) and we can measure it as a time series, then the memory timescale can be quantified by the time lag at which the autocorrelation of such a time series falls below its e-fold — or we can apply other methods like Hybrid Stochastic-Deterministic Model suggested by McColl et al. (2019), which is also used by Farmani et al. (2024); if the carrier of memory has no time series origin (like change in soil structure or pore size distribution), then other methods should be used for this quantification, such as the metrics used in paleopedology, I think.

Strength of Soil Memory: As used in the literature (e.g., Orth et al., 2013), this quantifies the strength of the drivers of Soil Memory. In this way, we can acknowledge and discuss that this memory is based on changes in atmospheric forcings, management factors, or soil properties and mechanisms. In the case of memory carriers with time series origin, it can be quantified by the value of autocorrelation at each time step from 1 (the day after the event) to the Memory Timescale.

Soil Legacy: This is the value of the impact of extreme events on the functioning, fluxes, and variables of the system after extreme events (which is probably of your interest when you talk about the strength of the feedback). For example, the change (positive or negative) in the fluxes of the system (soil respiration, $CO_2$ emission, etc.) in time steps after the occurrence of the extreme event. The legacy will certainly be stronger if we study it in close proximity to the event. As the temporal distance increases, the legacy decreases, and the impact is almost zero after a time corresponding to the time scale for memory. According to the literature, legacy can be quantified by comparing the state of the target variable or flux of the system at any time after an extreme event with the long-term average before the occurrence of that extreme event. Thus, it can be positive (e.g., an increase in $CO_2$ emissions after the extreme event) or negative (a decrease in carbon storage after the extreme event).

Best,
Mehdi Rahmati
Agrosphere Institute IBG-3,
Forschungszentrum Jülich GmbH,
Jülich, Germany

References:
Aliku, O. and Oshunsanya, S. O.: Assessment of the SOILWAT model for predicting soil hydro-physical characteristics in three agro-ecological zones in Nigeria, International Soil and Water Conservation Research, 6, 131-142, 2018.
Andrews, C. M. and Bradford, J. B.: SOILWAT: A Mechanistic Ecohydrological Model for Ecosystem Classification and Prediction, World Conference on Natural Resource Modeling. , 2016.
Caldwell, M. M., Dawson, T. E., and Richards, J. H.: Hydraulic lift: consequences of water efflux from the roots of plants, Oecologia, 113, 151-161, 1998.
Dawson, T. E.: Hydraulic lift and water use by plants: implications for water balance, performance and plant-plant interactions, Oecologia, 95, 565-574, 1993.
Farmani, M. A., Behrangi, A., Gupta, A., Tavakoly, A., Geheran, M., and Niu, G.-Y.: What Are the Key Soil Hydrological Processes to Control Soil Moisture Memory?, EGUsphere, 2024, 1-28, 2024.
Hagemann, S. and Stacke, T.: Impact of the soil hydrology scheme on simulated soil moisture memory, Climate Dynamics, 44, 1731-1750, 2015.
Jackson, R. B., Sperry, J. S., and Dawson, T. E.: Root water uptake and transport: using physiological processes in global predictions, Trends in plant science, 5, 482-488, 2000.
McColl, K. A., He, Q., Lu, H., and Entekhabi, D.: Short-Term and Long-Term Surface Soil Moisture Memory Time Scales Are Spatially Anticorrelated at Global Scales, Journal of Hydrometeorology, 20, 1165-1182, 2019.
Orth, R., Koster, R. D., and Seneviratne, S. I.: Inferring Soil Moisture Memory from Streamflow Observations Using a Simple Water Balance Model, Journal of Hydrometeorology, 14, 1773-1790, 2013.
Orth, R. and Seneviratne, S. I.: Propagation of soil moisture memory to streamflow and evapotranspiration in Europe, Hydrology and Earth System Sciences, 17, 3895-3911, 2013.
Oyeogbe, A. and Oluwasemire, K.: Evaluation of SOILWAT model for predicting soil water characteristics in southwestern Nigeria, International Journal of soil science, 8, 58, 2013.

Rahmati, M., Amelung, W., Brogi, C., Dari, J., Flammini, A., Bogena, H., Brocca, L., Chen, H., Groh, J., Koster, R. D., McColl, K. A., Montzka, C., Moradi, S., Rahi, A., Sharghi S., F., and Vereecken, H.: Soil Moisture Memory: State-Of-The-Art and the Way Forward, Reviews of Geophysics, 62, e2023RG000828, 2024.

Rahmati, M., Or, D., Amelung, W., Bauke, S. L., Bol, R., Franssen, H. J. H., Montzka, C., Vanderborght, J., and Vereecken, H.: Soil is a living archive of the Earth system, Nature Reviews Earth & Environment, 4, 421-423, 2023.

**R: Thanks for the comments. We have deleted the sentence "**These methods provide insights into the magnitudes of water and energy flux exchanges between land surface and atmosphere, indicating that shorter SMM durations can lead to more intense feedback and larger flux exchanges."

In the Introduction Section, we did not really summarize the mechanisms of SMM emergency. So, after reading your review paper. We have revised the Introduction to reflect we have learned:
"A recent review on SMM identified soil properties and processes as an important controlling factor of SMM in addition to atmospheric forcings and land use and management for future studies to examine the fundamental mechanisms of SMM emergence (Rahmati et al., 2024). Based on the works of McColl et al. (2019) and He et al. (2023), this study aims to examine the impacts of soil hydrological processes and soil hydraulics on SMM. The current LSMs may not be enough to address the uncertainties of SMM estinmates for incomplete representations of key hydrological processes controlling SMM and uncertainties in soil hydraulic parameters (Rahmati et al., 2024). As such, we use a version of Noah-MP with advanced hydrological representations of preferential flow, surface ponding, runoff of surface ponded water (infilration excess runoff), and lateral infiltration, etc. (Niu et al., 2024). We aim to optimize the soil hydraulics within the model by evaluating various parametrizations of those by Brooks and Corey (1964) and Van-Genuchten (1980), preferential flow, and surface ponding depth. Our analysis investigates the impact of these configurations on soil moisture consistency across different ET regimes and drainage, so it provides insight into physical processes affecting SMM. By comparing SMM produced by various settings of Noah-MP with SMAP Level 3 data and ISMN observations from 2015 to 2019 over the CONUS, we seek to identify key processes and soil hydraulic schemes controlling SMM and thus provide guidance for future developments of LSMs (e.g., reduce the prevalent SMM overestimations in LSMs)."

---

## Author Response (AR2)

Third Reviewer:
The authors have appropriately addressed my pervious comments, and I suggest to accept this paper after addressing my following two concerns.

1. For the results presented in Figures 3-6, how the authors match the Noah-MP results with SMAP results given the fact that the sampling frequency of SMAP data is one per three days. I think relevant description should be provided in the manuscript. The authors should also note that the sampling frequency will largely affect the computation of $\tau$L as given by Shellito et al. 2016.

R: We thank the reviewer for the thoughtful comment. To address the concern about how Noah-MP results were matched with SMAP observations, we clarify in the manuscript (Section 2.2.2) that the Noah-MP soil moisture data were selected to correspond to SMAP observation times. Additionally, the SMAP L3 soil moisture data were resampled to achieve a consistent sampling frequency of one observation every three days at each pixel, as previously stated:

We also note that the sampling frequency can significantly influence the computation of $\tau_L$, as highlighted by Shellito et al. (2016). To mitigate this potential impact, we ensured temporal alignment between Noah-MP outputs and SMAP observations, maintaining a consistent 3-day sampling interval.

We also added the following into paper:
"To ensure comparability, Noah-MP soil moisture data were selected to correspond to the SMAP observation times. This alignment minimizes potential biases introduced by temporal differences and facilitates a consistent analysis of soil moisture memory. It is important to note that the sampling frequency, as highlighted by Shellito et al. (2016), can significantly influence the computation of $\tau\_L$. This potential impact was mitigated in this study by aligning Noah-MP data with SMAP observation times and maintaining a consistent sampling frequency of one observation every three days, ensuring a reliable basis for analyzing soil moisture memory."

2. Considering the impact of sampling frequency, it's also interesting to know how the authors derive these results for rootzone for matching those analyses made for surface layer?

R: We thank the reviewer for this insightful comment. To address the concern regarding how root-zone results were derived to match analyses made for the surface layer, we clarify in the manuscript that both Noah-MP outputs and ISMN data were resampled to ensure temporal consistency with SMAP surface-layer observation times. Specifically, Noah-MP outputs for the root zone were sampled at the same temporal intervals as SMAP, and the ISMN data, which served as the benchmark for root-zone analyses, were also resampled to match SMAP observation times. This approach ensures alignment in sampling frequency across datasets, minimizing biases introduced by differing temporal resolutions and allowing for a consistent comparison between surface-layer and root-zone soil moisture memory analyses.

We also add the following into 2.2.3 section of the paper:

"For root-zone analyses, Noah-MP outputs were sampled to ensure temporal consistency with SMAP surface-layer observation times. Similarly, ISMN data were resampled to match the SMAP observation times, ensuring alignment in sampling frequency across all datasets used as benchmarks for root-zone soil moisture memory."